# Integration, coincidence detection and resonance in networks of spiking neurons expressing Gamma oscillations and asynchronous states

Eduarda Susin *, Alain Destexhe

Institute of Neuroscience (NeuroPSI), Paris-Saclay University, Centre National de la Recherche Scientifique (CNRS), Gif-sur-Yvette, France

* eduardadsusin@gmail.com

**Data Availability Statement:** Codes written in support of this publication are publicly available at ModelDB: http://modeldb.yale.edu/267039.

## Abstract

Gamma oscillations are widely seen in the awake and sleeping cerebral cortex, but the exact role of these oscillations is still debated. Here, we used biophysical models to examine how Gamma oscillations may participate to the processing of afferent stimuli. We constructed conductance-based network models of Gamma oscillations, based on different cell types found in cerebral cortex. The models were adjusted to extracellular unit recordings in humans, where Gamma oscillations always coexist with the asynchronous firing mode. We considered three different mechanisms to generate Gamma, first a mechanism based on the interaction between pyramidal neurons and interneurons (PING), second a mechanism in which Gamma is generated by interneuron networks (ING) and third, a mechanism which relies on Gamma oscillations generated by pacemaker *chattering* neurons (CHING). We find that all three mechanisms generate features consistent with human recordings, but that the ING mechanism is most consistent with the firing rate change inside Gamma bursts seen in the human data. We next evaluated the responsiveness and resonant properties of these networks, contrasting Gamma oscillations with the asynchronous mode. We find that for both slowly-varying stimuli and precisely-timed stimuli, the responsiveness is generally lower during Gamma compared to asynchronous states, while resonant properties are similar around the Gamma band. We could not find conditions where Gamma oscillations were more responsive. We therefore predict that asynchronous states provide the highest responsiveness to external stimuli, while Gamma oscillations tend to overall diminish responsiveness.

## Author summary

In the awake and attentive brain, the activity of neurons is typically asynchronous and irregular. It also occasionally displays oscillations in the Gamma frequency range (30–90 Hz), which are believed to be involved in information processing. Here, we use computational models to investigate how brain circuits generate oscillations in a manner

**Funding:** A.D was supported by the Centre National de la Recherche Scientifique (CNRS) and the European Community (Human Brain Project, H2020-785907). E.S. acknowledges a PhD fellowship from the École des Neurosciences de Paris (ENP) and from the Fondation pour la Recherche Médicale (FRM) - grant FDT202012010566 - and the financial support from La Fondation des Treilles. The funders had no role in study design, data collection and analysis, decision to publish, or preparation of the manuscript. CNRS: www.cnrs.fr Human Brain Project: www.humanbrainproject.eu ENP: www. paris-neuroscience.fr FRM: www.frm.org La Fondation des Treilles: https://www.les-treilles. com/la-recherche/le-prix-jeune-chercheur/.

**Competing interests:** The authors have declared that no competing interests exist.

consistent with microelectrode recordings in humans. We then study how these networks respond to external input, comparing asynchronous and oscillatory states. This is tested according to several paradigms, an *integrative mode*, where slowly varying inputs are progressively integrated, a *coincidence detection mode*, where brief inputs are processed according to the phase of the oscillations, and a *resonance mode* where the network is probed with oscillatory inputs. Surprisingly, we find that in all cases, the presence of Gamma oscillations tends to diminish the responsiveness to external inputs. We discuss possible implications of this responsiveness decrease on information processing and propose new directions for further exploration.

## Introduction

Gamma oscillations appear in many brain states and brain regions [1] and are detectable mostly from the local field potential (LFP) as oscillations in the 30–90 Hz frequency range. During sensory responses, oscillations in this frequency range were initially proposed to serve as a mechanism for coordination of neural activity among cells coding for different aspects of the same stimulus [2–5]. Strengthening of synaptic input due to temporal summation led to the hypothesis that Gamma synchrony was necessary to effectively transmit specific sets of information across cortical networks in the very noisy conditions in which the brain operates. This concept was later expanded by proposing that synchronous Gamma also engages inhibition in target networks. Phase-locked inhibition creates strong suppression around the excitatory drive and creates windows of low and high neuronal excitability. Such observations led to hypotheses that Gamma oscillations are important for information processing and coding. The most popular theories are the Binding-by-synchronization Hypothesis [4, 5], the Phase Coding Theory [6, 7], the Communication Through Coherence Theory [8, 9] and Communication through Resonance Theory [10].

An alternative hypothesis, instead of relying on oscillations for efficient cortical communication, posits that *desynchronized states* are optimal for the transfer of signals between cortical networks [11, 12]. Desynchronized states, called *Asynchronous-Irregular* (AI) [13] because of its features, are characterized in cortical cells in vivo by irregular firing with very weak correlations and stationary global activity [14–18]. This type of activity can be modeled by networks with balanced excitatory and inhibitory inputs [19].

In the present work, we aim at testing these two discrepant points of view using computational models. We take advantage of previously published electrophysilogical data, measured extracellularly in human temporal cortex [20, 21], to characterize the behavior of individual neurons during Gamma oscillations in resting awake states, and to compare such experimental features to spiking neural networks generating Gamma. We exploit different network structures to investigate three well-known mechanisms of Gamma generation [22–27]: either by the exclusive interaction between inhibitory neurons [*Interneuron Gamma* (ING)] or by the interaction of inhibitory and excitatory neurons via *Pyramidal-Interneuron Gamma* (PING) or via *Chattering Induced Gamma* (CHING). First we compare to what degree each mechanism can reproduce the observed experimental features of human Gamma oscillations and what are the specificities of each mechanism, in the way neurons behave during Gamma. Subsequently, we examine network responsiveness due to three types of stimulus: Gaussian slowly-varying inputs (*integration mode*), precisely-timed Gaussian inputs (*coincidence detection mode*) and a sinusoidal varying Poissonian input (*resonance*).

## Materials and methods

### Neuron and network models

Each of the three networks developed in this work uses the *Adaptive Exponential Integrate-And-Fire Model* (Adex) [28] for its neural units. In this model, each neuron $i$ is described by its membrane potential $V_i$, which evolves according to the following equations:

$$
\begin{aligned}
C\frac{dV_i(t)}{dt} &= -g_L(V_i - E_L) + g_L\Delta exp\left[\frac{(V_i(t) - V_{th})}{\Delta}\right] - w_i(t) - I_{Syn_i}(t) \\[1em]
I_{Syn_i}(t) &= g_{E_i}(t)(V_i(t) - E_E) + g_{I_i}(t)(V_i(t) - E_I) \\[1em]
\tau_{E,I}\frac{dg_{E,I_i}(t)}{dt} &= -g_{E,I_i}(t) + Q_{E,I_i}\sum_k \delta(t - t_k) \\[1em]
\tau_{w_i}\frac{dw_i(t)}{dt} &= a(V_i(t) - E_L) - w_i(t) + b\sum_j \delta(t - t_j)
\end{aligned}
\tag{1}
$$

where $C$ is the membrane capacitance, $g_L$ is the leakage conductance, $E_L$ is the leaky membrane potential, $V_{th}$ is the effective threshold and $\Delta$ is the threshold slope factor. The synaptic current ($I_{Syn_i}$ (t)) received from other neurons to neuron $i$ is taken into account as conductance based: every time a presynaptic neuron spikes at time $t_k$, the excitatory ($g_{E_i}$) or the inhibitory ($g_{I_i}$) synaptic conductance increase by a discrete amount $Q_E$ or $Q_I$ (excitatory or inhibitory synaptic strength), depending on the nature of the presynaptic neuron. Synaptic conductances subsequently decay exponentially with a time constant $\tau_E$ or $\tau_I$. $E_E$ and $E_I$ are the reversal potential of excitatory ($E_E$) and inhibitory ($E_I$) synapses. The $\Sigma_k$ runs over all the presynaptic excitatory or inhibitory neurons spike times. During the simulations, the equation characterizing the membrane potential $V_i$ is numerically integrated until a spike is generated. Formally this happens when $V_i$ grows rapidly toward infinity. In practice, the spiking time is defined as the moment in which $V_i$ reaches a certain threshold ($V_{th}$). When $V_i = V_{th}$ the membrane potential is reset to $V_{rest}$, which is kept constant until the end of the refractory period $T_{ref}$. After the refractory period the equations start being integrated again. The adaptation current is described by the variable $w_i$. It increases by an amount $b$ every time neuron $i$ emits a spike at times $t_j$ and decays exponentially with time scale $\tau_w$. The parameter $a$ indicates the subthreshold adaptation.

Three types of cells were used in our models: Regular Spiking Cells (RS), Chattering Cells (Ch) and Fast Spiking Cells (FS). The cell specific activities are displayed in Fig 1 and their parameters are indicated in Table 1.

Each of the three developed networks are composed of N = 25000 neurons, 80% excitatory and 20% of inhibitory. All neurons are connected randomly. Additionally to recurrent connections, each neuron receive an external drive (noise). This noise was implemented as $N_{Ext}$ = 20000 independent and identically distributed excitatory Poissonian spike trains with a spiking frequency $\mu_{Ext}$, being sent to the network with a 2% probability of connection. These spike trains were computed inside of the synaptic current term $I_{syn}(t)$, by means of a discontinuous increase of the excitatory synaptic conductance $g_E$ by an amount $Q_{Ext}$ (at every spike time). This type of implementation adds to the network a low degree of correlation, since some neurons share the same drive. Nevertheless, this extra correlation does not affect our results, which kept being qualitatively the same when a drive with no correlations was applied. The

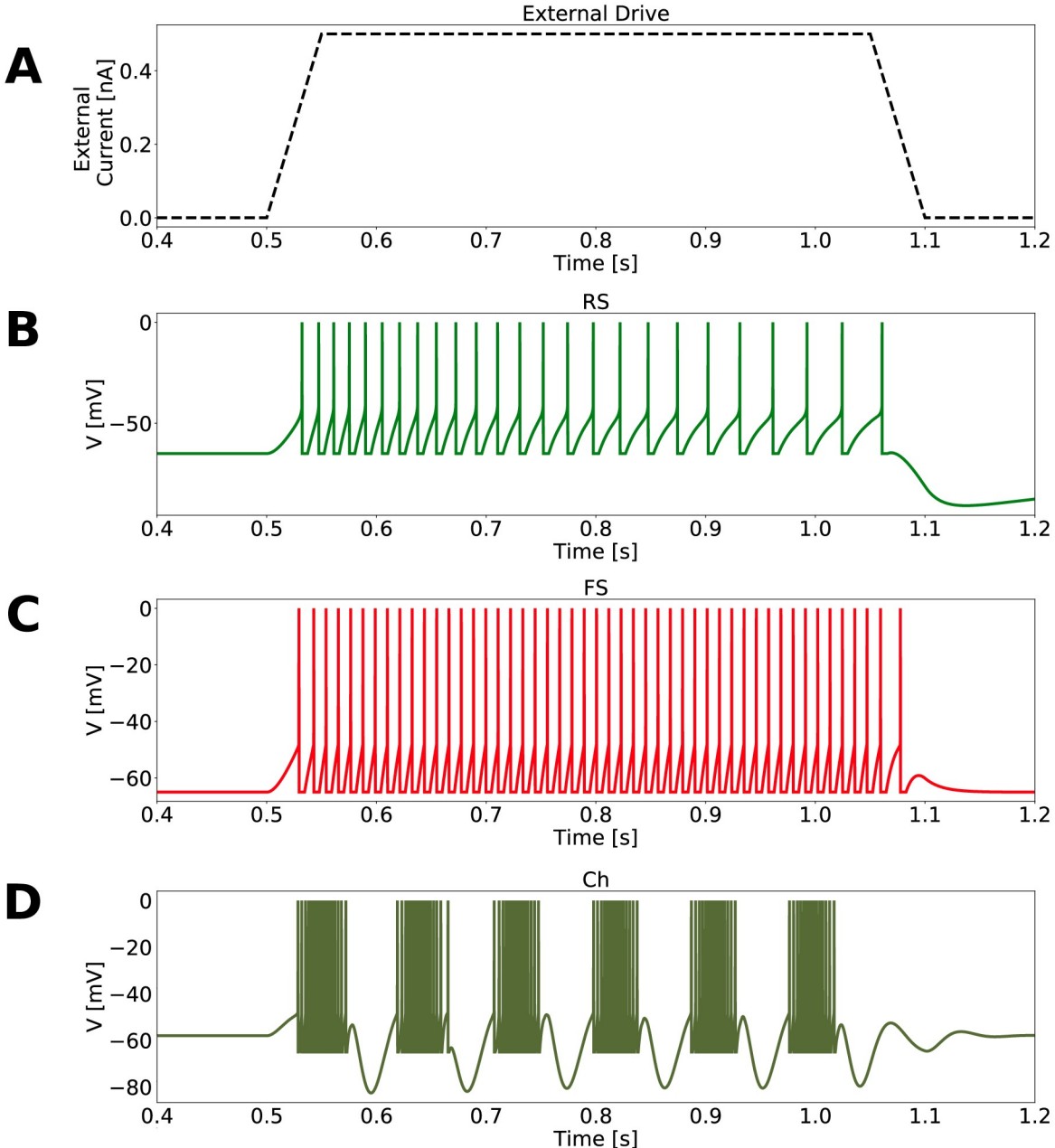

**Fig 1. Neuronal response to an external current.** A: External drive fluctuation. External current, in each neuron, varied from 0 to 0.5 nA in a linear way, was kept constant for 500 ms, subsequently decreasing to 0 nA in a linear way. B: Isolated RS cell in response to the external drive presented in A. C: Isolated FS cell in response to the external drive presented in A. D: Activity of one Ch cell, in a network exclusively composed of 1000 Ch cells connected randomly with a probability of 2%.

patterns of connection and neuron type composition of each network model, as well as the specific values of Poissonian stimulation ($\mu_{Ext}$ and $Q_{Ext}$), are described bellow.

- **PING Network**: It is composed of 25000 Adex neurons (20000 excitatory Regular Spiking and 5000 inhibitory Fast Spiking cells). All neurons are connected randomly with a probability of connection of 2%. All synapses are delayed by a time delay of 1.5 ms. The synaptic excitatory (inhibitory) time scales are $\tau_E = 1.5$ ms ($\tau_I = 7.5$ ms), with synaptic strengths of

**Table 1. Specific neuron model parameters.**

| Parameter | RS | FS | Ch |
|---|---|---|---|
| $V_{th}$ | -40 mV | -47.5 mV | -47.5 mV |
| $\Delta$ | 2 mV | 0.5 mV | 0.5 mV |
| $T_{ref}$ | 5 ms | 5 ms | 1 ms |
| $\tau_w$ | 500 ms | 500 ms | 50 ms |
| $a$ | 4 nS | 0 nS | 80 nS |
| $b$ | 20 pA | 0 pA | 150 pS |
| $C$ | 150 pF | 150 pF | 150 pF |
| $g_L$ | 10 nS | 10 nS | 10 nS |
| $E_L$ | -65 mV | -65 mV | -58 mV |
| $E_E$ | 0 mV | 0 mV | 0 mV |
| $E_I$ | −80 mV | −80 mV | −80 mV |
| $V_{rest}$ | -65 mV | -65 mV | -65 mV |

$Q_E$ = 5 nS ($Q_I$ = 3.34 nS). Synaptic time scales were chosen accordingly to the parameter search indicated in S1 Fig. For Gamma activity, the network was stimulated with an external noise of $\mu_{Ext}$ = 3 Hz and $Q_{Ext}$ = 4 nS. For an activity similar to an Asynchronous and Irregular activity (AI-like), the network was stimulated with an external noise of $\mu_{Ext}$ = 2 Hz and $Q_{Ext}$ = 4 nS.

- **Asynchronous and Irregular (AI) Network**: The *AI Network* was used in this work as one of the building blocks for the ING and the CHING Network. It is composed of 25000 neurons (20000 excitatory Regular Spiking and 5000 inhibitory Fast Spiking). All neurons are connected randomly with a probability of connection of 2%. All synapses have synaptic strengths of $Q_E$ = 1 nS or $Q_I$ = 5 nS, and are delayed by a time delay of 1.5 ms. This network, independently of the strength of the the external noise, can not generate Gamma rhythms. This is the case because the chosen synaptic excitatory and inhibitory time scales ($\tau_E = \tau_I = 5$ ms) are in a region of the parameter space in which the regime is asynchronous and irregular. See S1 Fig. Because of this feature, the *AI Network* was used as a control to study network responsiveness (see Results section).

- **Gamma Network**: The *Gamma Network* was used in this work as one of the building blocks for the ING Network. It is composed of 1000 inhibitory Fast Spiking neurons, highly connected between each other. All neurons are connected randomly with a probability of connection of 60%. All synapses have synaptic strengths of $Q_I$ = 5 nS and synaptic time constant of $\tau_I$ = 5 ms, and are delayed by a time delay of 1.5 ms. This network is capable of generating oscillations by its own due to the exclusive presence of inhibitory neurons excited by an external drive [29, 30]. Low oscillation frequencies in the Gamma range ($\approx$70 Hz) are possible thanks to the high connectivity patterns used (60%). S2 Fig displays the parameter space of network connectivity vs. inhibitory synaptic strengths for this network. The parameters chosen in our simulations (p = 60% and $Q_I$ = 5 nS) are indicated.

- **ING Network**: The *ING Network* is constructed as a mixture of *AI network* with the *Gamma Network*. It is composed of 25000 neurons: 20000 RS and 4000 FS from the *AI network* plus 1000 FS neurons from the *Gamma Network*. The Fast Spiking neurons in the original *AI network* and the ones in the *Gamma Network* share all the same parameters of FS cells in Table 1. The only difference among them is their pattern of connectivity. To make it clear, we call as FS2, the FS neurons that were part of the *Gamma Network*, and we keep calling as FS the ones that were part of the *AI Network*. In the *ING Network*, FS2 cells send and receive

random connections to RS neurons with a probability of 15%, FS2 cells send random connections to FS neurons with a probability of 15% while FS cells send random connections to FS2 neurons with a probability of 3%. This combination of the *Gamma network* with the *AI Network* allows the oscillation frequency to slow down further, reaching $\approx$ 55 Hz. All synapses have synaptic strengths of $Q_E$ = 1 nS or $Q_I$ = 5 nS and synaptic time scales of $\tau_E = \tau_I = 5$ ms. Synapses are delayed by a time of 1.5 ms. For Gamma activity the network was stimulated with an external noise of $\mu_{Ext}$ = 3 Hz, while for Asynchronous and Irregular activity, the network was stimulated with an external noise of $\mu_{Ext}$ = 2 Hz. The external noise used had a synaptic strength of $Q_{Ext}$ = 0.9 nS.

- **CHING Network**: The *CHING Network* is constructed the same way as the *AI network*, with the difference that 5% of the RS cells were replaced by Chattering Cells (Ch). This way, the *CHING Network* is composed of 25000 neurons: 19000 RS, 1000 Ch and 5000 FS. All cells in the network are randomly connected to each other with a probability of 2%. All synapses have synaptic time scales of $\tau_E = \tau_I = 5$ ms and are delayed by a time delay of 1.5 ms. Excitatory synapses have synaptic strengths of $Q_E$ = 1 nS, while inhibitory synapses from FS cells to Ch or to RS have synaptic strengths of $Q_I$ = 7 nS. Synapses from FS to FS have synaptic strengths of $Q_I$ = 5 nS. The network receives external noise with synaptic strength of $Q_{Ext}$ = 1 nS in excitatory cells (RS and Ch) and $Q_{Ext}$ = 0.75 nS in FS cells. For Gamma, external noise of $\mu_{Ext}$ = 2 was used, while for Asynchronous and Irregular activity, $\mu_{Ext}$ = 1 Hz.

## Simulations

All neural networks were constructed using Brian2 simulator [31]. All equations were numerically integrated using Euler Methods and dt = 0.1 ms as integration time step. The codes for each one of the three developed networks are available at ModelDB platform: http://modeldb.yale.edu/267039.

## LFP model

To model the LFP generated by each of the three developed networks, we used a recent method developed by [32]. This approach calculates the contribution of individual neurons to the LFP by means of the convolution of individual neuron spike trains (generated by the networks) with a phenomenological Kernel $\mathcal{K}$, which had its parameters fitted from unitary LFPs (the LFP generated by a single axon, *uLFP*) measured experimentally [32]. Each neuron spike train is convoluted with a particular Kernel $\mathcal{K}^p$ that depends on the particular neuron position $\vec{x_p}$ in a 2-D space.

$$\mathcal{K}^p(\vec{x}, t) = A(\vec{x}) exp[-(t - t_{pick})^2/(2\sigma^2)]$$

$$t_{pick} = t_0 + d + |\vec{x} - \vec{x_p}|/v_a \qquad (2)$$

$$A(\vec{x}) = A_0 exp[-|\vec{x} - \vec{x_p}|/\lambda]$$

in which $\sigma$ is the standard deviation in time, $t_{pick}$ is the peak time of the *uLFP*, $t_0$ is the time of the spike of a particular cell $p$, $d$ is a constant delay, $v_a$ is the axonal speed, and $|\vec{x} - \vec{x_p}|$ is the distance between the position of particular cell ($\vec{x_p}$) and the position of the electrode ($\vec{x}$). $A(\vec{x})$ gives the space-dependent amplitude, in which $A_0$ is the maximal amplitude, and $\lambda$ is the space constant of the decay. These parameters were estimated separately for excitatory and

inhibitory contributions ($\mathcal{K}_E^p$ and $\mathcal{K}_I^p$) [32, 33]. The LFP, at a particular electrode position $\vec{x}$, is given by the sum of all individual neuron contributions:

$$LFP(\vec{x}, t) = \tag{3}$$

$$\sum_p \int \mathcal{K}_E^p(\vec{x}, t - \tau) \left( \sum_j \delta(t - t_p^j) \right) d\tau + \sum_p \int \mathcal{K}_I^p(\vec{x}, t - \tau) \left( \sum_j \delta(t - t_p^j) \right) d\tau \tag{4}$$

In which $\sum_p$ runs over neurons $p$, and $\sum_j$ runs over all spike times of neuron $p$. To be able to apply this method to our simulations (which don't presume any neuronal localization in space), we randomly displaced the network neurons in 2-D grid, assuming that the electrode was displaced on its center and was measuring the LFP in the same layer as neuronal soma. The program code of the kernel method is available in ModelDB (http://modeldb.yale.edu/266508), using python 3 or the *hoc* language of NEURON.

## Detection of Gamma rhythms and Gamma phase

In both, experimental and simulated signals, Gamma rhythms were detected by means of the Hilbert transform of the band-filtered LFP. The identification of Gamma bursts was done separately for each electrode. We considered as Gamma bursts periods in which the amplitude of Hilbert Transform envelope (absolute value) differed from the mean, by at least 2 standard deviations for the experimental data, and by at least 1 standard deviation for the numerical ones, for a minimum duration of 3 Gamma cycles. This criteria were not enough to identify all Gamma bursts (some Gamma bursts were ignored). On the other hand, no false positives were included in the analysis. All the Gamma bursts automatically identified by the algorithm were individually confirmed visually. The oscillation phase was acquired using the angle of the imaginary part of the transform. The LFP was band-pass filtered in the band of 30–50 Hz (unless indicated otherwise). To band-pass the LFP signals, we used a FIR (Finite Impulse Response) filter using the Kaiser window method with a 60 dB stop-band attenuation and a 5Hz width from pass to stop transition [34]. To implement the filter we used the following functions from the Python-based ecosystem *Scipy*: *signal.kaiserord*, *signal.lfilter* and *signal.firwin* [35].

## Spike-LFP phase-locking

Every time a Gamma period was identified, in both experimental and simulated signals, the spiking times of each neuron was stored and compared to the Gamma rhythm phase. This information allowed the construction of the phase distribution of each neuron. For the experimental data, considering that the identification of Gamma bursts was done separately for each electrode, neurons measured in particular electrode, had their phases and firing rates analyzed exclusively with respect to the rhythm measured in this electrode. Neuron phases were calculated from $-\pi$ to $\pi$. In this way neurons with negative phases should be interpreted as spiking preferentially before than neurons with positive phases. The phase distribution of each neuron was tested for circular uniformity using a Bonferroni-corrected Rayleigh test [36, 37]. A neuron was considered phase-locked if we could reject circular uniformity at P < 0.01. See S3 Fig. Neurons that spiked less then 5 times inside Gamma bursts, or neurons whose electrode measured less then 1 second of Gamma, in the respective data segment, were classified as *inconclusive*.

## Firing rate change

The average firing rate of each neuron outside Gamma bursts ($f_{out}$) was computed based in the total time, excluding the activity inside Gamma bursts and their duration. In accordance, the

average firing rate inside Gamma bursts ($f_\gamma$) was calculated based on the total Gamma duration and the activity occurring exclusively inside Gamma bursts. A neuron was considered to increase its firing significantly if the observed number of spikes in the measured time was higher than the *percent point function* of a 95% Interval of Confidence of a Poissonian distribution with average firing rate $f_{out}$. Cells that had firing rates smaller then 0.1 Hz or cells whose electrode measured less then 1 second of Gamma bursts, in the respective data segment, were classified as *inconclusive*. See S4 Fig.

## Responsiveness

The level of *responsiveness* ($R$) of a network, due to a stimulus ($S$) in a time window of duration $T$, is defined as the difference between the total number of spikes generated by the whole network due to a stimulus ($N^S_{spikes}$) and the total number of spikes generated in the absence of the stimulus ($N_{spikes}$), normalized by the network size (total number of neurons $N_n$) and the duration of the time window $T$.

$$R = \frac{N^S_{spikes} - N_{spikes}}{T N_n} \tag{5}$$

## Phase-dependent responsiveness

The *Phase-dependent responsiveness* of a network $R(\theta)$, in a time window of duration $T$, due to a stimulus $S$ presented to the network in a particular phase $\theta$ of the Gamma cycle, is defined as the difference between the total number of spikes generated by the whole network due to a stimulus at the $\theta$ phase, $N^S_{spikes}(\theta)$, and the total number of spikes generated in the absence of the stimulus at the $\theta$ phase, $N_{spikes}(\theta)$, normalized by the network size (total number of neurons $N_n$) and the time window $T$.

$$R(\theta) = \frac{N^S_{spikes}(\theta) - N_{spikes}(\theta)}{T N_n} \tag{6}$$

## Human recordings

In one epileptic patient with intractable seizures, 10x10 Neuroprobe silicon multielectrode arrays (400-$\mu$m inter-electrode separation, 1 mm electrode length, Blackrock Microsystems) were implanted in the middle temporal gyrus (layers II/III). Electrodes were implanted in regions expected to be removed, and after the monitoring session, the implant area was excised. The patient consented to the procedure, which was approved by the Massachusetts General Hospital Institutional Review Board in accordance with the ethical standards of the Declaration of Helsinki. This data set have already been published previously [20, 21]. Neurons could be classified through clustering based on the spike shape and functional interactions (determined using cross-correlograms) [20, 38] as Regular Spiking Cell (RS), putative excitatory, and Fast Spiking Cells (FS), putative inhibitory. From 81 electrodes, 91 neurons could be detected: 23 FS and 68 RS.

## Results

We first analyze Gamma oscillations from human recordings, then examine network models of Gamma oscillations and compare them to the experimental data. Finally, we examine the

responsiveness and resonant properties of these networks, comparing Gamma and asynchronous states.

## Human recordings analysis

In this paper, aiming to constrain our computational models to observed experimental features, we extend the human data analysis performed in [20, 21], focusing on awake states. The data was acquired extracellularly in patients suffering of intractable epilepsy, who had multi-electrode arrays implanted during therapeutic procedures. The arrays registered simultaneously local field potential (LFP) and unit activity. We considered here one patient for which the recording was very stable, and in which several periods of wakefulness could be analyzed.

In each electrode, Gamma rhythms were identified and neural activity was characterized with respect to the Gamma cycles. Fig 2A illustrates a specific instant in which Gamma bursts were observed in most of the electrodes (spiking activity and the respective electrode band-filtered LFP are shown). Gamma rhythms were determined through the Hilbert transform of the filtered LFP (30–50 Hz). Fig 2B and 2C give an example of how Gamma is detected and how neural phase with respect to the oscillation is extracted (see *Detection of Gamma rhythms and Gamma phase* in Materials and methods Section). The data were acquired during the night. Five awake periods could be recorded, having a mean duration of 27 minutes, containing on average 13 seconds of Gamma (Fig 2D). During these periods the patient was in a resting awake condition.

In accordance with other studies, the spiking activity during Gamma bursts was observed to be very irregular and close to a Poissonian process, with a spiking frequency much smaller than the population frequency [21, 39–41]. Moreover, conformable to [21], on average, only 4% of RS cells and 17% of FS cells were Phase-Locked (Fig 2E), with RS cells having a phase preference later in the cycle than the FS cells (see S5 Fig). Furthermore, by measuring the firing rate change of each cell inside and outside Gamma bursts (Fig 2F), we encountered on average 47% of FS cells that increased their firing inside Gamma bursts, while only 17% of RS cells did. These observations suggest that Gamma oscillations modulate spiking activity in two manners: by means of firing rate increase and by defining time windows were some neurons are more likely to spike (phase-locking).

Contrary to the intuition that all neurons in a network generating Gamma would be *participating* to the rhythm, this analysis indicates that, only a small percentage of neurons has its activity modulated by the oscillation (either by phase-locking or by firing rate increase). We call this group of neurons as *Gamma participating* cells.

To better characterize the *non-participation* to Gamma rhythms, we followed each cell in each of the 5 waking periods present in the recordings, searching for behavioral changes. We observed that in different data segments, different groups of neurons were identified to participate to Gamma, indicating that the group of *Gamma participating* cells varies with time (see S6 Fig). Furthermore, cells that were classified as phase-locked in different data segments, had their preferred phase changed from one recording to the other (see cells 65 and 22 in S5 Fig). We called this feature as *dynamical phase preference*. Fig 3 indicates the individual cell *behavior consistency*, that is, how frequently a cell keeps being identified to a certain behavior: either being phase-locked or to have its firing rate changed inside Gamma bursts in a particular data segment. Stacked bars of Fig 3A and 3B indicate a color-coded behavior distribution of individual neurons, inside of the 5 data segments, with respect to firing rate change and phase-locking respectively. Neurons are ordered in a way in which inhibitory cells are displayed in the beginning. Red neuron indexes stand for FS cells and green neuron indexes stand for RS cells. Fig 3C and 3D depict the distribution among all recorded neurons of each behavior (C:

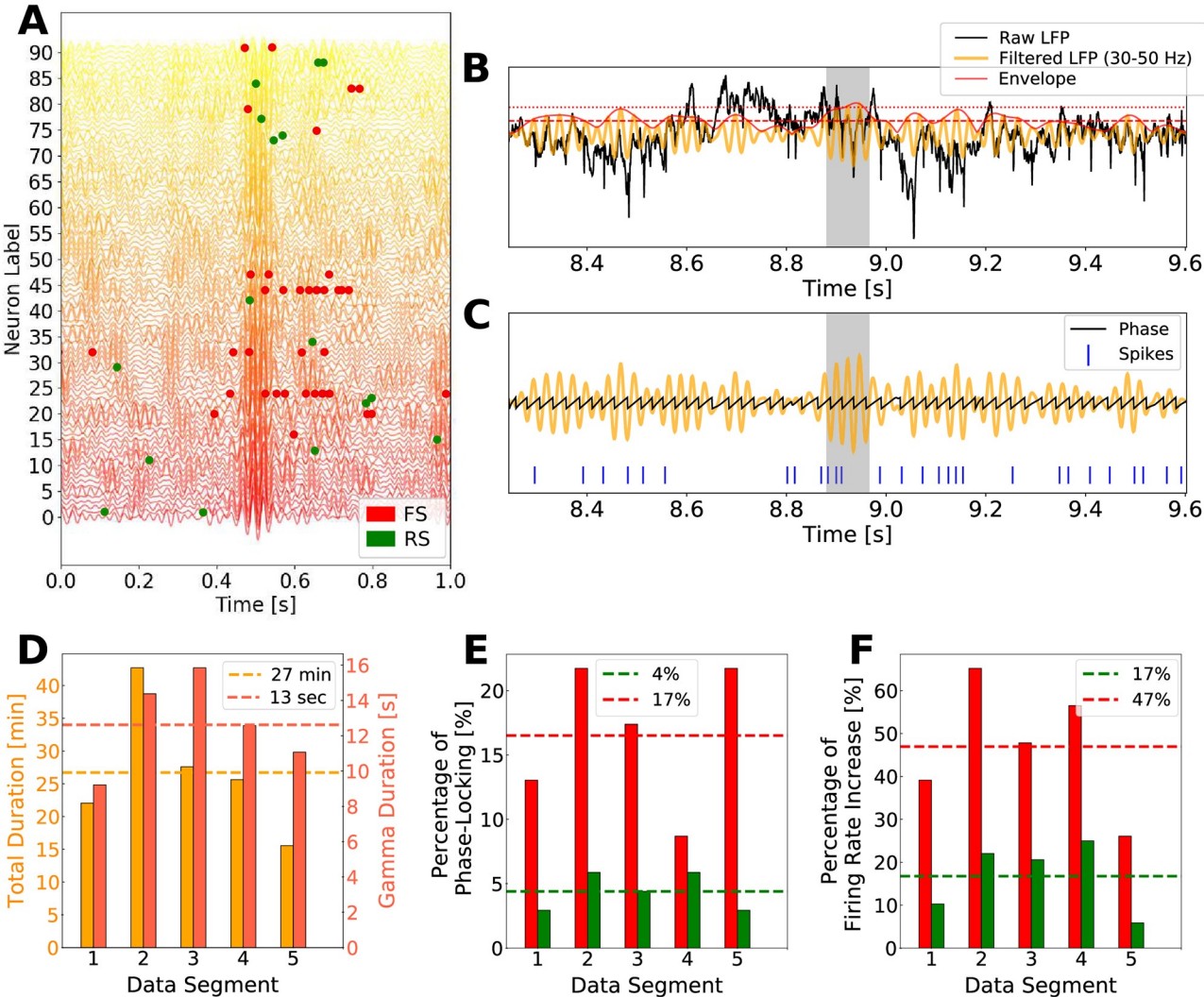

**Fig 2. Human electrophysiological data.** A: Simultaneously recorded LFP and multi-units activity. The Filtered LFP (30–50 Hz) of the 81 electrodes are shown together with the spiking times of 91 neurons. Some neurons were recorded by the same electrode, which had its LFP duplicated in the figure. The identification of Gamma bursts was done separately for each electrode. This way, neurons measured in a particular electrode, had their phases and firing rates analyzed exclusively with respect to the rhythm measured in its respective electrode. Spikes of Fast Spiking (FS) neurons, presumably inhibitory, are shown in red, and spikes from Regular Spiking (RS) neurons, presumably excitatory, are shown in green. B: Gamma periods detection. Raw LFP (black), band-pass filtered LFP (yellow) and Hilbert Transform Envelope (red) are shown. Gamma bursts were detected by means of the deviation from the average of the Hilbert Transform envelope (dashed red line) of at least 2 SDs (dotted red line), with a minimum duration of 3 Gamma cycles. The gray shaded region indicates one example of identified Gamma burst. C: Oscillation Phase extraction. The oscillation phases were obtained by the angle of the imaginary part of the Hilbert Transform. The phase distributions of each neuron were computed based on the oscillation phases where each neuron spiked. D: Data organization. Five awake periods could be recorded during one night. Each period had a different total time duration (yellow bars in minutes) and a different average duration of total Gamma occurrences (orange bars in seconds). Since each electrode was analyzed individually, the average indicated in the bars is the average among all the electrodes in the respective segment. E: Percentage of neurons identified as phase-locked in each data segment. The average amount of Phase-locked neurons in the five data segments was of 4% in RS and 17% in FS. RS neurons are shown in green and FS neuron in red. F: Percentage of neurons that increased their firing during Gamma, in each data segment. The average amount neurons in the five data segments which increased their firing during Gamma was of 17% in RS and 47% in FS. Same color scheme as in E.

Firing Rate Increase, D: Phase-Locking). A behavior consistency of zero denotes that the indicated percentage of neurons never presented that behavior, while a behavior consistency of 5 denotes that the indicated percentage of neurons presented that behavior in all 5 data segments. FS cells tended to participate of Gamma bursts with higher consistency than RS cells.

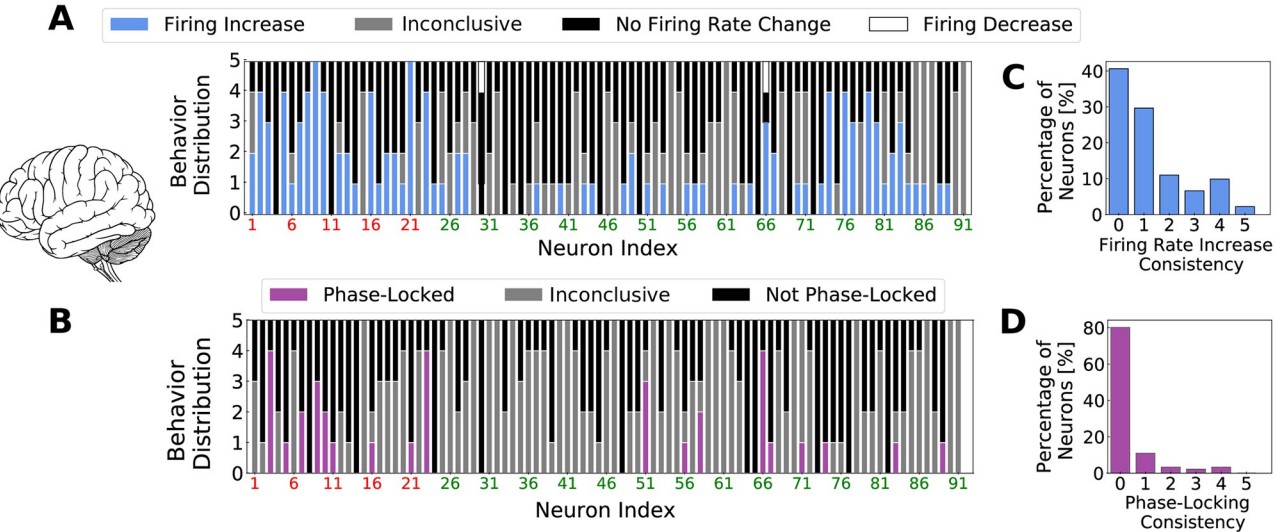

**Fig 3. Individual neural behavior consistency on human recordings.** Stacked bars indicating the color-coded distribution, inside of the 5 data segments, of individual neural behavior relative to firing rate change (*A*) and phase-locking (*B*). Neurons are ordered in a way in which inhibitory neurons are displayed in the beginning of the graph. Red neuron indexes stand for FS cells and green neuron indexes stand for RS cells. Items *C* and *D* indicate respectively the statistics of the consistency indexes among the recorded neurons for Firing Rate Increase and Phase-Locking.

While 34.8% of FS increased their firing inside Gamma bursts in at least 4 of the 5 data segments, only 4.4% of RS cells did the same. Moreover 8.7% of FS cells kept being phase-locked in at least 4 data segments, in comparison to only 1.5% in RS population (see S7 Fig). Likewise, we call the reader's attention to the significant number of cells that never increase their firing rate inside Gamma bursts (Fig 3C, ≈ 40% of the recorded neurons) and to the significant number of cells that never presented phase-locking (Fig 3D, ≈ 80% of the recorded neurons). The behavior of individual cells during Gamma is quantified in S8 Fig.

Furthermore, another important aspect to be acknowledge is a possible correlation between high firing rate cells (inside Gamma bursts) and those cells that show higher phase locking. Nonetheless, the human data set used in this study is too small to be able to arrive to any conclusion. In our analysis cells with high firing rates were observed to be not phase-locked (or phase-locked), the same way as cells with lower firing rates were observed to phase-locked (or not phase-locked). See S9 Fig. The same is true if we try do drive conclusions about the co-occurrences of firing rate increase and phase-locking (see S6 Fig).

In summary our analysis shows that, during Gamma bursts, only a small percentage of the recorded neurons participate of the rhythm. This participation takes place in two ways: phase-locking and/or firing rate increase. FS cells presented significant higher level of phase-locking and firing rate increase in comparison to RS cells. Likewise the level of consistency behavior were also more marked in FS cells than RS cells. Our analysis further indicates that, the group of *Gamma participating* cells changes with time as well as their phase-preference.

## Network models of Gamma oscillations

Gamma oscillations have been extensively modeled in the literature with different neuronal models and networks structures [23, 42]. The low and irregular firing rates observed during Gamma oscillations have been reproduced in recurrent networks of spiking neurons [13, 30, 43–45] by means of strong recurrent inhibition and strong noise (due to external inputs and/or due to synaptic disorder). Networks displaying this type of activity are known to be in the

*firing rate regime* [30]; in contrast to models fully synchronized, in which neurons behave as periodic oscillators. In this last regime, known as an *spike-to-spike regime*, neurons spike at every cycle (or once every two cycles), with an average firing rate close to the frequency of oscillatory network activity [46–54]

It is well established, experimentally and theoretically, that inhibition plays a crucial role in generating Gamma rhythms [21, 23, 29, 42, 55–59]. Nonetheless, it is still controversial [22–25] whether Gamma oscillations are generated by the exclusively interaction among inhibitory neurons [*Interneuron Gamma* (ING)] or via the interaction of inhibitory and excitatory neurons [*Pyramidal-Interneuron Gamma* (PING)]. Furthermore, a third mechanism, less explored in the literature, relies on the presence of pacemaker excitatory cells known as *Chattering neurons* [26, 27]. We named this third mechanism as *Chattering Induced Gamma* (CHING).

To compare to what degree each of three previously mentioned mechanisms can reproduce the observed experimental features, and what are the consequences of each mechanism, we constructed three neural networks working in the *firing rate regime*, adapted to generate Gamma by means of ING, PING or CHING. Network and neuronal parameters were chosen in a way to allow each model to reproduce experimental features as well as possible, with physiologically plausible firing rates and membrane conductance distributions (see S10 and S11 Figs). We call the reader's attention to the fact that, while networks with a structure similar to our *PING Network* have been largely used in the literature, the structures of *ING* and *CHING Networks* were developed exclusively for this study.

Like in previous works [60], all three networks are capable of generating spontaneous Gamma bursts. These Gamma bursts are controlled by fluctuations of recurrent drive generated by the network dynamics, which for this reason occur irregularly and in an unpredictable fashion. However, more predictable Gamma bursts can be obtained by increasing the external drive (in all three networks). Fig 4 shows the behavior of the three networks when a fluctuation on the Poissonian input generates Gamma, mimicking the Gamma bursts observed experimentally. Note however that, outside of Gamma bursts (low input amplitude), the networks do not necessarily display a pure AI state: all three networks display reminiscent low-amplitude oscillations. In all cases, the firing dynamics remained irregular and with low synchrony, so we called them *AI-like states*.

We next performed on the network models an equivalent analysis as in the human data recordings. Each cell was followed in 5 different simulations containing on average 13 seconds of Gamma bursts (same duration as in the experimental recordings, mimicking the five experimental data segments) and statistical tests to identify phase-locking and firing rate changes were performed. Fig 5A, 5B and 5C display respectively the quantification of behavior consistency for *PING*, *ING* and *CHING* Networks. Accordingly to the unit recordings [21], the cells were generally more depolarized and increased their firing during Gamma. On the other hand, within the three models, only the *ING Network* (Fig 5B*c*) is capable of describing the appropriate amount of neurons that increase their firing in different data segments, during Gamma. The *PING* and *CHING* networks predict an over-estimation of this number. The presence of a sub-population of highly connected inhibitory neurons, capable of generating Gamma rhythms by their own (see *Neuron and Network Models: Gamma Network* in Materials and methods Section), allows the *ING Network* to provide a compensation for external excitatory fluctuations: whenever there is an augmentation of input in the network (generating Gamma), there is in addition a concomitant augmentation of inhibition thanks to the FS2 population.

In comparison to the experimental data analysis performed previously, all three models are capable of correctly describing the frequency of re-occurrence of phase-locking inside of a group of neurons in different data segments. That is, all three models predict the same the

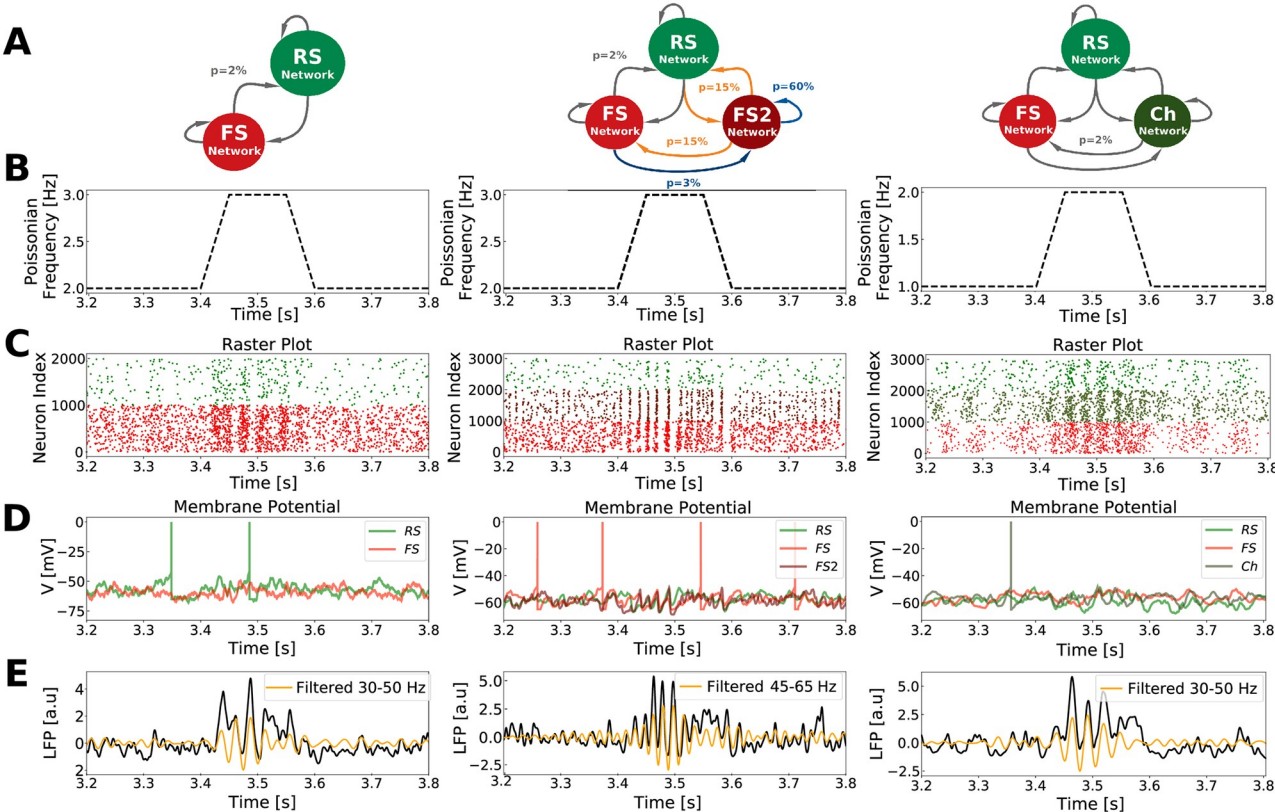

**Fig 4. Neural activity of different Gamma generation mechanisms networks.** *PING Network* (left), *ING Network* (middle) and *CHING Network* (right). A: Scheme of each network structure and pattern of connectivity. B: External Poissonian noise fluctuation generating Gamma bursts. C: Raster plot of network activity inside and outside Gamma bursts. Only 1000 neurons of each cell type are shown. D: Membrane potential activity of randomly picked neurons of each type. Pay attention to the well defined subthreshold oscillation exclusively present in the *ING Network*. E: Simulated LFP (raw—in black) and its filtered version (yellow).

same intensity of *phase-locking consistency* as the one observed on the human recordings (Fig 3D). On the other hand, regardless of the mechanisms of Gamma generation, all networks predict an over estimated phase-locking level (total number of phase-locked neurons per data segment) (see S12 Fig). With respect to the human data set, the *PING* and *ING* networks predict a comparable level of phase-locking in the excitatory population but an exaggerated level in the inhibitory population. In contrast, the *CHING Network* predicts a comparable level of phase-locking in the inhibitory population but an exaggerated level in the excitatory one. Side by side, the *CHING Network* is the one that still captures the best the level of phase-locking in both populations (excitatory and inhibitory).

The right prediction of phase-locking consistency can be explained by the type of activity regime in which each network works: the *fluctuation-driven regime*. Since this regime allows neurons to spike with low firing rates in an irregular fashion, participating of the global Gamma oscillation only in certain cycles due to the subthreshold randomness. Nonetheless, the over-estimation of phase-locking level, indicates that the simple fact of being in the *fluctuation-driven regime* is not enough to capture all levels of description. We hypothesize that the network structure play a key role in the way neurons behave during oscillations. Fig 5 illustrates how different network structures (different connectivities in the ING Network or different neuron types in the CHING Network) influence network activity.

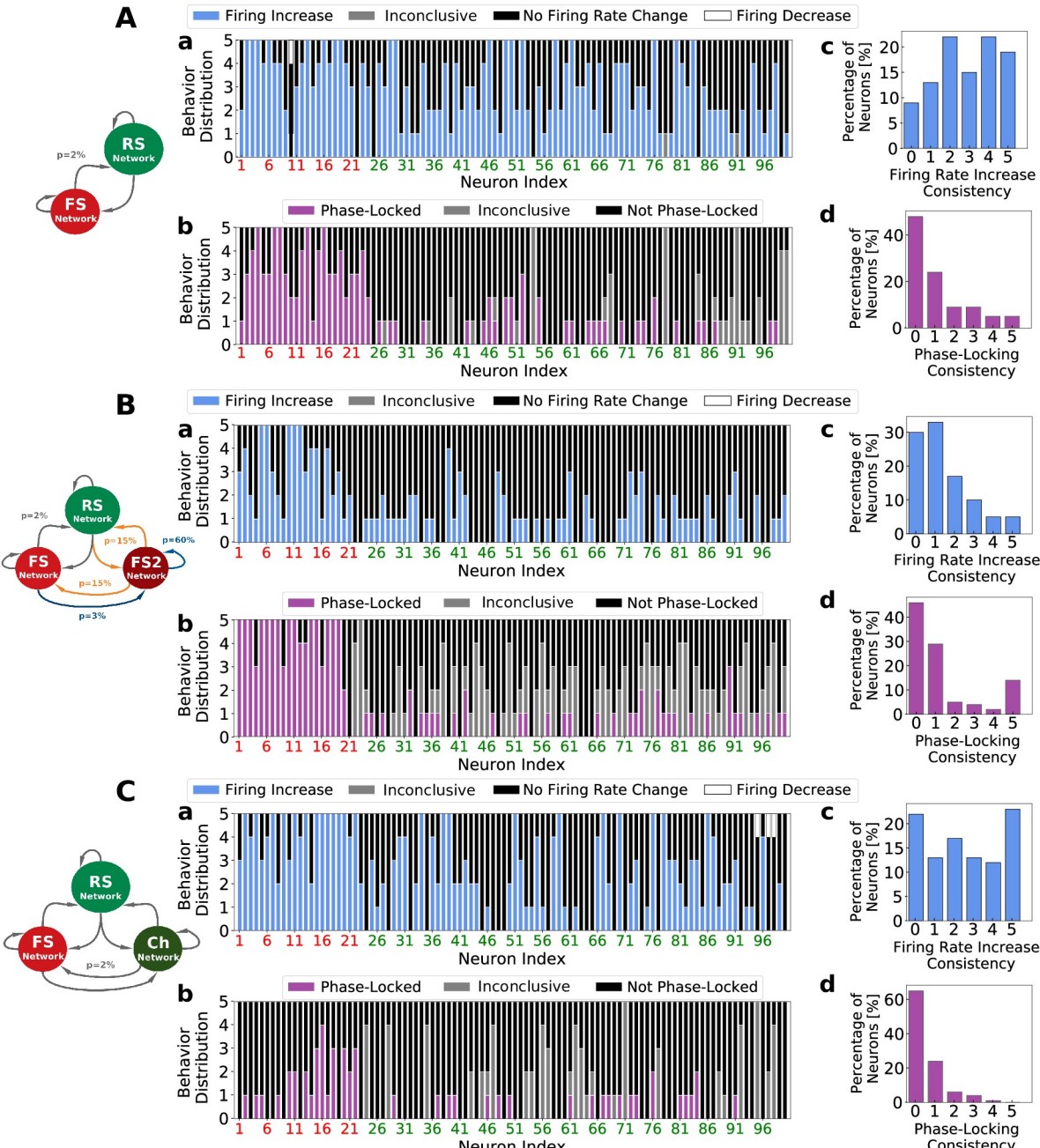

**Fig 5. Individual neural behavior consistency in computational models.** A: *PING Network*. B: *ING Network*. C: *CHING Network*. Same analysis and color codes used in Fig 3. To mimic the five experimental independent data segments in the Human data recordings (Fig 3) on the network models, five simulations (per model) were performed, containing on average the same amount of total Gamma bursts duration as in the experimental data (13 seconds). In addition, to match the number of recorded neurons in the experimental data, in the models a subset of 100 randomly picked neurons were selected in each case.

In the presented human recordings, inhibitory neurons tended to spike earlier in the cycle than excitatory neurons. Fig 6 shows the phase preference with respect to the Gamma cycle of all the neurons considered phase-locked in the human data recordings (Fig 6A) and in each of the three developed networks (Fig 6B, 6C and 6D). The *ING* and *CHING* networks predict the

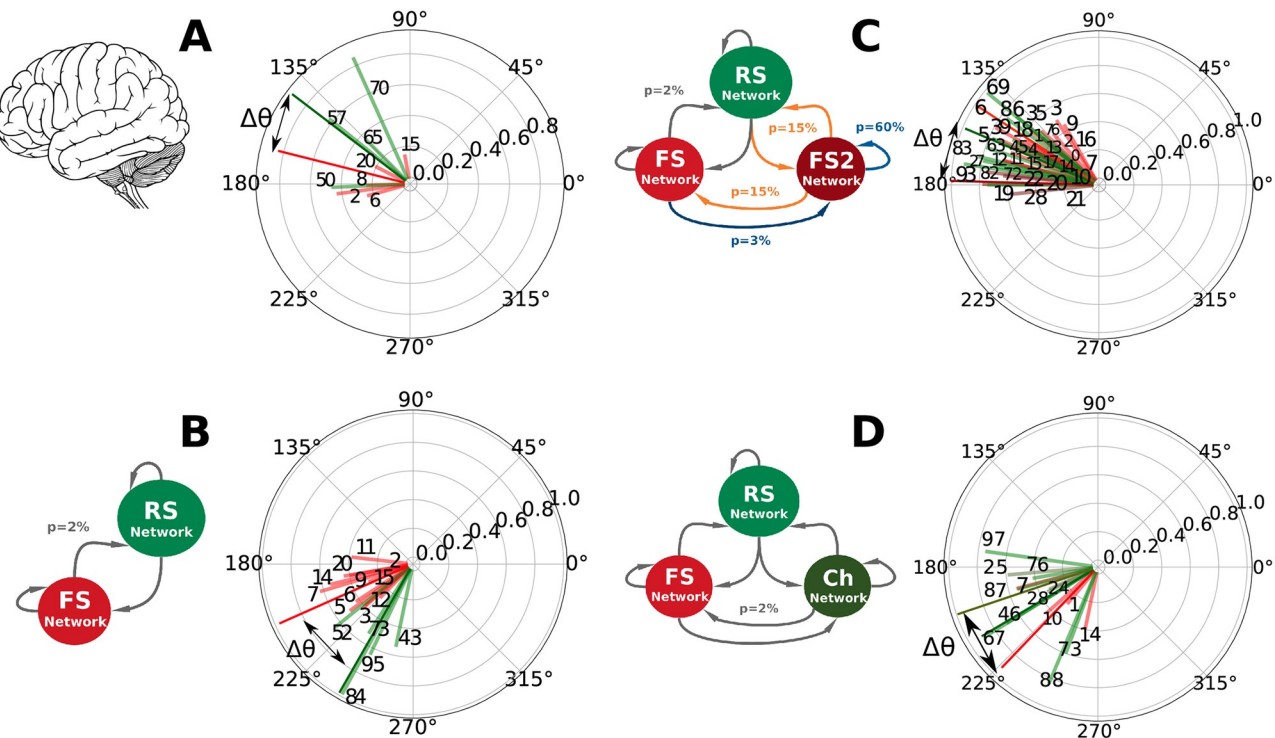

**Fig 6. Phase preference of phase-locked cells.** A: Human Data (Data segment 2). B: *PING Network* Data. C: *ING Network* Data. D: *CHING Network* Data. The preferred phases of each phase-locked cell are displayed in polar graph representation. Note that, since phases were calculated from $-\pi$ to $\pi$ (see *Spike-LFP phase-locking* in Materials and methods Section), these polar graphs should be interpreted clockwise with time. The vector size gives a measure of the phase distribution of each cell. Big amplitude vectors indicate very concentrated distributions while small amplitude vectors indicate less concentrated ones (see S3 Fig). The color of each vector encodes the type of the cell of whom it represents the phase: red (FS), dark red (FS2), green (RS) and dark green (Ch). Cell number IDs are indicated. Dark colored vectors indicate the average phase among each neuron type and $\Delta\theta$ the phase difference among them. Data segment 2 presented 43 minutes of recordings, containing 14 seconds of Gamma activity.

same relationship as observed in the human recordings (inhibition preceding excitation) while the *PING Network* predicts the opposite. Moreover, in the same way as the human data set (S5 Fig), cells that were classified as phase-locked, have their preferred phase changed from one simulation to other (dynamical phase preference). We argue that this feature is also a consequence of the *fluctuation-driven regime*.

The phase relationship between excitation and inhibition is an important aspect to be discussed, since it has been suggested to be a marker of the type of Gamma generation mechanism [25]. It has been shown theoretically by [45] that, in models composed of conductance based neurons (neurons that include non-linear spike generation mechanisms on their equations) the spiking order of excitatory and inhibitory populations depends exclusively on single-cell characteristics. Based on their analysis, when the $I_{AMPA}/I_{GABA}$ ratio is the same in excitatory and inhibitory neurons, excitatory cells tend to follow the inhibitory ones in most of the physiologically plausible parameter space. On the other hand, when the ratio of excitation to inhibition is weaker in excitatory cells than in inhibitory ones, excitatory cells tend to precede inhibitory neurons [30, 45]. In our simulations, the only network in which this theory can be directly applied (because of the network structure) is the *PING Network*, in which the $I_{AMPA}/I_{GABA}$ ratio in excitatory cells is weaker than in inhibitory cells. Interesting discussions about neural properties and population phase-differences can also be found on [61, 62].

Concluding this section, we showed that network models working in the *firing rate regime*, regardless of the mechanism of Gamma generation, can reproduce qualitatively some of the

most important features of experimental neural activity during Gamma: phase-locking consistency and dynamical phase preference. On the other hand, all models predict an overestimation of the phase-locking levels. Additionally, only the *ING Network* model was capable of describing a reasonable level of firing rate increase inside Gamma bursts, as found in the human recordings. We advocate that just the simple fact of being in the fluctuation-driven regime is not enough to capture all levels of description of Gamma oscillations, and hypothesize that the network structure play a key role in the way neurons behave during oscillations.

Considering that the different types of spontaneous activity exhibited by the three presented models could greatly influence how the network processes external input, we have investigated this issue of responsiveness to external input in the next section.

### Responsiveness and resonance during Gamma oscillations

**Responsiveness.**   The way information is encoded and processed in the brain is still a largely investigated enigma. Several ways of encoding information have been considered, such as firing rates [63, 64], pairwise correlations [65, 66], spike pattern irregularity [67–70] and spike packets [71], among others. In particular, two main theories have been dominating the debate: *Temporal Coding* in which individual neurons encode information by means of precise spike timings (working as coincidence detectors), and the *Rate Coding* in which neurons encode information by means of changes in their spike rates (working as temporal integrators). Regardless of the encoded strategy used to encode information, the way the network is capable of responding to a certain stimulus is of prime importance. To identify how Gamma rhythms change the response properties of a network to an external stimulus with respect to AI, in this section we applied two protocols, investigating the effect of Gamma in both, the *coincidence detection mode* and in the *integration mode* [72, 73].

In the *integration mode* protocol, we compared how each of the three developed models responded to slowly-varying inputs (occurring in a time window much bigger than the Gamma period). In this protocol, each network received Poissonian drive (spikes from an external network) with firing rates varying in time, in a Gaussian manner, both during Gamma and AI-like states. The applied Gaussian inputs had a standard deviation of 50 ms, allowing the stimulus to interact with different Gamma cycles. Several amplitudes of slowly-varying Gaussian were tested, and the *responsiveness* of excitatory and inhibitory populations were measured separately. *Responsiveness* (see Eq 5) was defined as the difference between the total number of spikes (in a time window of duration T) generated by the whole network in the presence and in the absence of the stimulus (normalized by the network size and the time window duration T).

Fig 7 shows the *responsiveness* of the *PING Network*, the *ING Network* and the *CHING Network*, when the *integration mode* protocol was applied. To be able to measure the real impact of Gamma oscillations on network responsiveness, we used as a control the responsiveness curves from the AI-Network, in which no oscillation is generated, independently of the level of external drive. See *Neuron and Network Models: AI Network* in Materials and methods Section and S13 Fig. All models, regardless of the mechanism of Gamma generation, were less responsive during Gamma bursts in comparison with their baseline responsiveness during AI-like states. Furthermore, the responsiveness of a network in a real AI-state (gray curve generated by the AI Network) is equal or higher to the AI-like responsiveness, in each of the networks, and is always higher then the Gamma state responsiveness in all cases. In addition, to further investigate this result, we examined the responsiveness of individual cells (S14 Fig). Due to the previous finding that only a restricted group of cells *participate* to Gamma, one could imagine that there could still be few cells (*Gamma participating* cells) that would be more responsive,

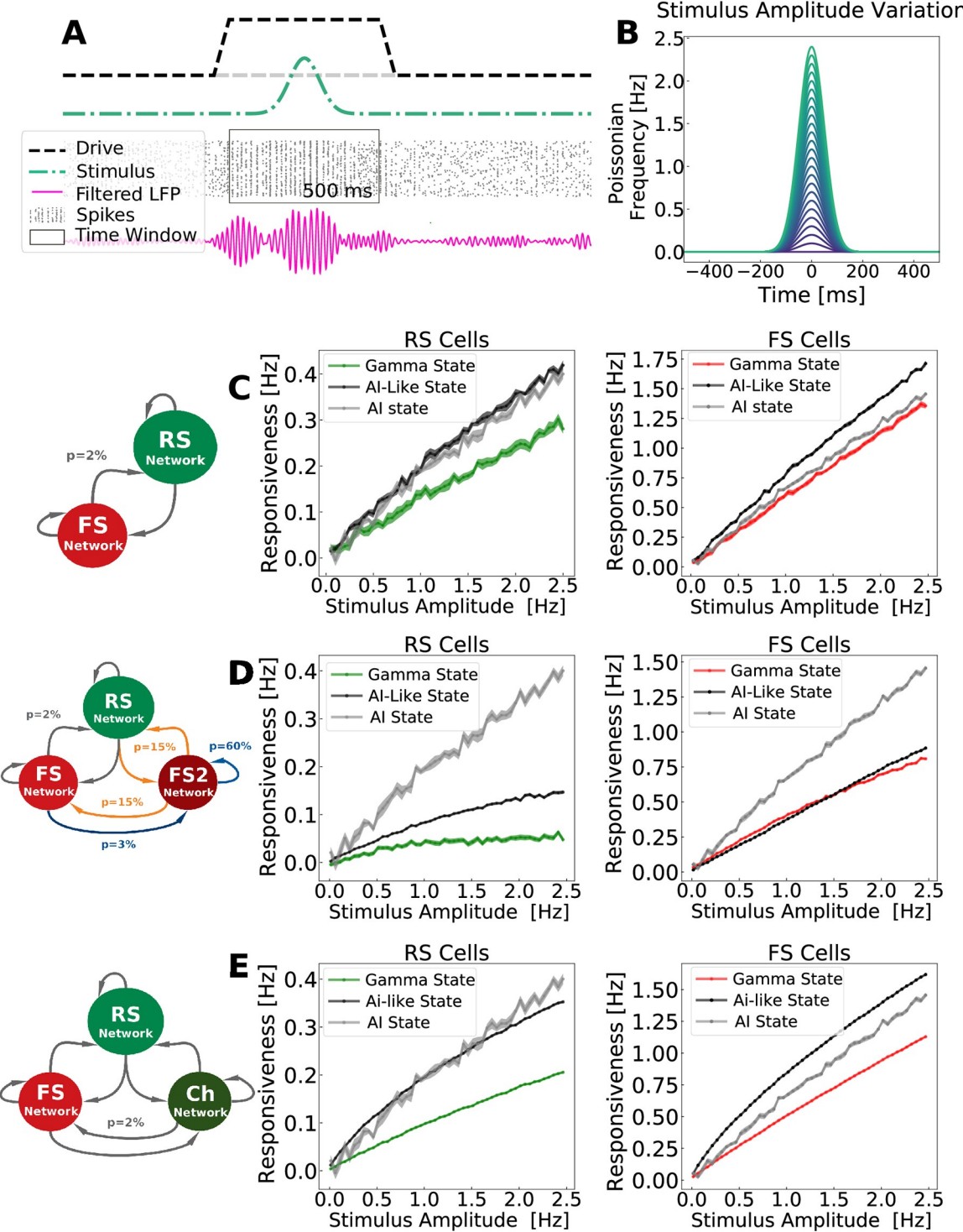

**Fig 7. Network responsiveness to a Gaussian input with varying amplitude.** The *responsiveness*, in different states (Gamma and AI-like), was measured in the three developed networks and compared to the responsiveness of the *AI Network* as a control. A: Responsiveness protocol scheme for Gamma state. A Gamma burst is generated due to fluctuations of the external drive (black dashed line). During the Gamma activity, a Gaussian input (green line) is applied. The total number of spikes due to the stimulus, in time window of 500 ms, is measured. To measure the total number of spikes in the absence of the stimulus, another drive fluctuation is created generating Gamma. The total number of spikes inside of a time window of 500 ms is measured again (this time, without the Gaussian input). Only the situation in response to a stimulus is depicted in the scheme. This procedure was repeated 100 times per each Gaussian amplitude input. To measure responsiveness in AI and AI-like states no drive fluctuation was applied (the black dashed line in the figure

was kept constant). B: Input Amplitude Variation. The stimulus consisted of a Gaussian fluctuation in the firing rate of the external drive. The Gaussian amplitude varied from 0.05 Hz to 2.5 Hz (step of 0.05 Hz) and had a standard deviation of 50 ms. Items C, D and E display respectively the responsiveness of the *PING Network*, the *ING Network* and the *CHING Network*, inside Gamma bursts (green for excitatory cells, red for inhibitory cells), and outside Gamma bursts—AI-like activity—(black for both types of cells). Every point corresponds to the average responsiveness measured in 100 simulations. Standard error of the mean are indicated by the shaded region around each curve. The responsiveness of the *AI-Network* was added as a control in each case (gray curve in C, D and E). To implement the responsiveness protocol the *AI-Network* received a constant drive with $\mu_{Ext}$ = 3 Hz, in addition to the Gaussian inputs. See S13 Fig.

while all others (*Gamma non-participating* cells) would be less responsive, leading to a yet overall less prominent responsiveness. Nonetheless, S14 Fig shows the contrary. All cells seem to follow the same decrease of responsiveness during Gamma oscillations, and we found no evidence that some subset of cells would be more responsive, for all amplitudes tested.

In the *coincidence detection mode* protocol, the responsiveness at different Gamma phases was measured. To do this, precisely-timed inputs (occurring in a time window much smaller than the Gamma period) were applied and related to Gamma cycles in each of the three developed networks. In this protocol the amplitude of the stimulation was kept constant, while the time of the application of the Gaussian stimulus changed with respect to the phase of the Gamma oscillation. This procedure allowed each network to be stimulated at different Gamma phases (see S15 Fig). Fig 8 indicates the network response of excitatory cells per Gamma phase, in different states: Gamma state (blue), AI-like (black) and AI-like modulated by a control external current oscillating at Gamma frequency (gray). All responses were normalized by the average response of AI-like states without external current modulation (black).

AI-like states, when modulated by an external oscillatory current, displayed, in all network models, preferred phases in which the network response was higher in comparison to the non-modulated AI-like state. This constitutes an important control, because the external current creates periods of higher and lower excitability in the network, which is translated in a phase-dependent response (as shown by the gray curves in Fig 8). Likewise, when generating Gamma, our models (PING and ING) demonstrate an equivalent type of phase-dependence response (even-tough with a narrow amplitude range). On the other hand, in agreement with the *integration mode* protocol, our simulations show that the responsiveness during Gamma states at all phases are less or equal to that during AI-like states.

**Resonance.**    In Physics, when dealing with an oscillatory system, one of the first features to be explored is its resonant properties. In general, *resonance* describes the phenomenon of increased amplitude in a system, that occurs due to the application of an oscillatory stimulus whose frequency is equal or close to the natural frequency of the system. It has been shown experimentally that this phenomenon can also be observed in inhibitory [56] and excitatory [74] neuronal populations. Furthermore, theoretical studies [75] have shown that resonance is a fundamental property of spiking networks composed of excitatory and inhibitory neurons. Resonance has also been proposed as a mechanism to gate neuronal signals [76] and to communicate information [10].

We tested the resonant properties of each of our networks in AI-like and Gamma states. In this protocol, each network received Poissonian drive with firing rates varying in time in a sinusoidal manner, with different frequencies (Fig 9A). Fig 9B, 9C and 9D depict, for each frequency and oscillation phase, the average number of spikes per RS neuron and time bin, during Gamma and AI-like states, for the *PING*, *ING* and *CHING* Networks. All values were normalized by the average firing inside of each state to exclude the state dependent firing rate level (which is higher on Gamma). To enhance the comprehension of the responsive properties of each network, a linear version of the color maps depicted in Fig 9 (amplitude vs. phase)

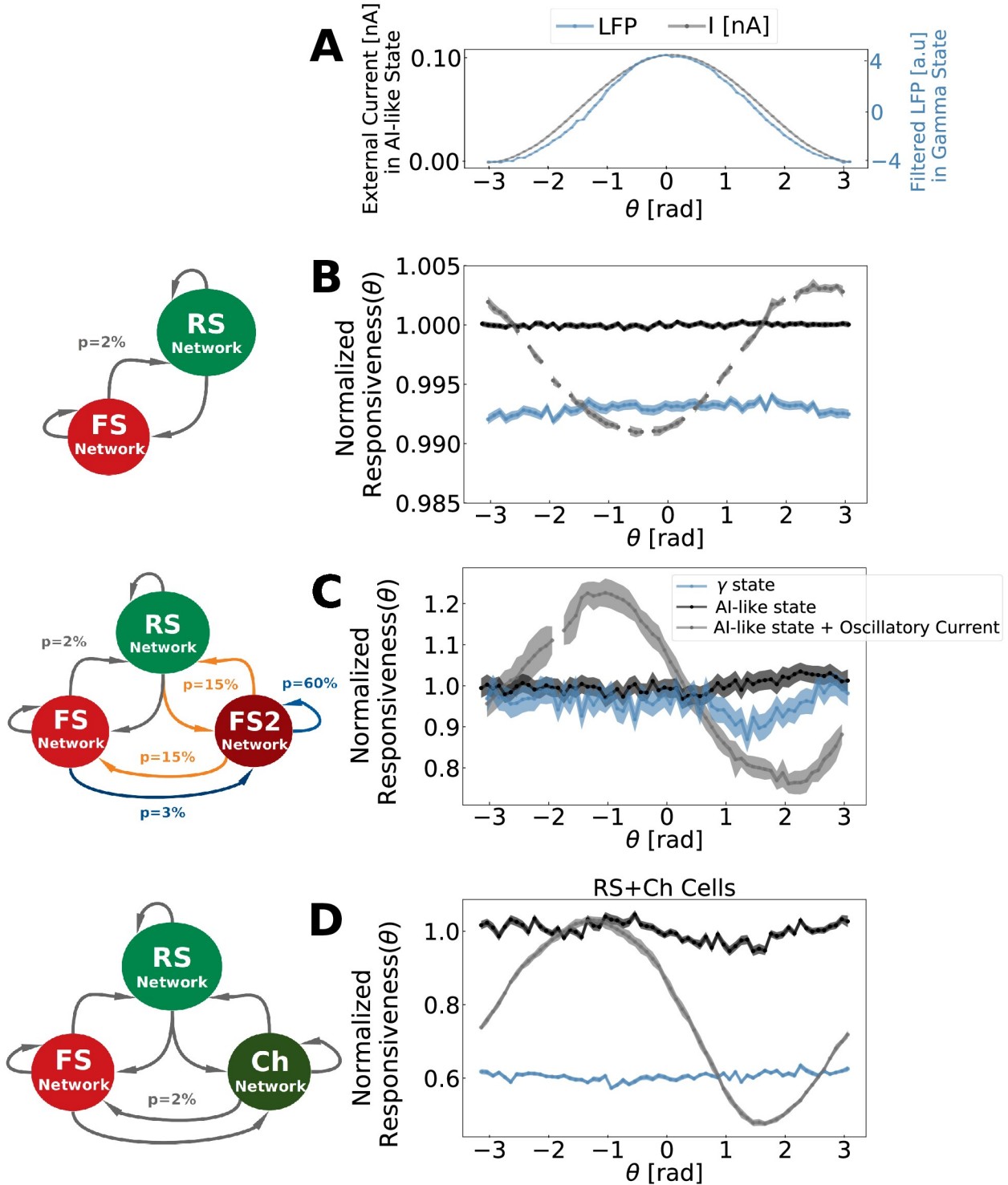

**Fig 8. Phase-dependent network response.** A: External oscillatory current applied at AI-like state as function of its oscillation phases (gray curve) and the filtered LFP measured during Gamma states as function of its oscillation phases (blue curve). All networks received a current oscillating from 0 to 0.1 nA in a sinusoidal manner with a Gamma frequency $F_\gamma$. To match the Gamma oscillation frequency generated by each network, the frequency of the external current applied to PING and CHING networks was $F_\gamma = 40$ Hz, while the one applied to ING network was $F_\gamma = 55$ Hz. The LFP depicted is the one from PING network. ING and CHING also displayed a similar LFP pattern. B: *PING Network* phase-dependent response C: *ING Network* phase-dependent response. D:*CHING Network* phase-dependent response. The phase-dependent network response was calculated according to Eq 6, in a time window of duration $T$ equal to one Gamma cycle (T = 25ms for the PING and CHING Networks and T = 18ms for

ING). Responses measured inside AI-like activity (outside Gamma bursts) are shown in black, and in gray when the networks received a supplementary oscillatory external current. Responses measured inside Gamma bursts are displayed in blue. All curves were normalized by the average response inside AI-like activity without external current modulation. Solid lines indicate the average, and the shaded region indicates the standard error of the mean. The curves were calculated based on the output of 12000 simulations (120 positions of the Gaussian stimulus in 100 numerical seeds for external Poissonian drive). The Gaussian stimulus used had an amplitude of 50 Hz and standard deviation of 1 ms.

is provided in S16 Fig. In addition, S17 Fig depicts the resonant properties in other cell types (FS, FS2 or Ch) during Gamma state for each one of the networks.

We observe that, in both AI-like and Gamma states, all models display resonant properties around the Gamma band, with the main difference in between these two states being a shift of the resonance frequency center. In this protocol we detect a similar level of responsiveness per phase (reflected in the measured number of spikes per time bin) in AI and Gamma, indicating that networks receiving oscillatory inputs have the same latent potential to resonate at Gamma ranges regardless if they are displaying AI or Gamma oscillations. One should note that each model presents its own particularities. While the PING network presents just a shift of the center frequency of resonance, the ING network presents an enlarged potential of resonance in AI (in addition to the frequency shift). During AI, the ING network presents an equal resonance in several bands other than Gamma. Moreover, when a Gamma oscillation is triggered in this network, this resonance is shrunk and becomes more concentrated in the Gamma band. The CHING network, on the other hand, presents a strong resonance in the 15–25 Hz frequency range during AI, while during Gamma this resonance is lost.

Concluding this section, we investigated three dynamical properties (Responsiveness, Phase-dependent-responsiveness and Resonance) in different states (AI-like and Gamma) of each of the three developed networks. We encounter that, regardless of Gamma generating mechanism (PING, ING or CHING), the network responsiveness, in both *coincidence detection* and *integrative* mode, is decreased at Gamma states with respect to AI. On the other hand, the resonant properties around the Gamma band in all networks did not change significantly from one state to the other. The main resonant properties changes between AI and Gamma states in each model were most prominent around other bands. The implications of these observations on the role Gamma rhythms in neural computations and information transfer will be discussed in the next section.

## Discussion

In this paper, we have examined the genesis and responsiveness of Gamma oscillations constrained by human recordings. We analyzed Gamma oscillations from previous studies [20, 21], where the recordings were stable, and in which RS and FS cells were discriminated. We compared the results of this analysis to conductance-based network models implementing three different mechanisms that were proposed for Gamma oscillations, PING, ING and CHING. We next examined these three networks with respect to their responsiveness and resonance to external inputs. We discuss these aspects below.

### Human data analysis

Compared to a previous analysis of the cellular correlates of Gamma oscillations [21], we confirm here the low level of cellular engagement and a greater participation of FS cells during Gamma, either through phase-locking or through firing rate increase. FS cells not only presented a higher percentage of phase-locking or firing rate increase during Gamma, but they also presented a more consistent behavior compared to RS cells which were much more variable. Our analysis further indicates that, the group of Gamma participating cells changes with

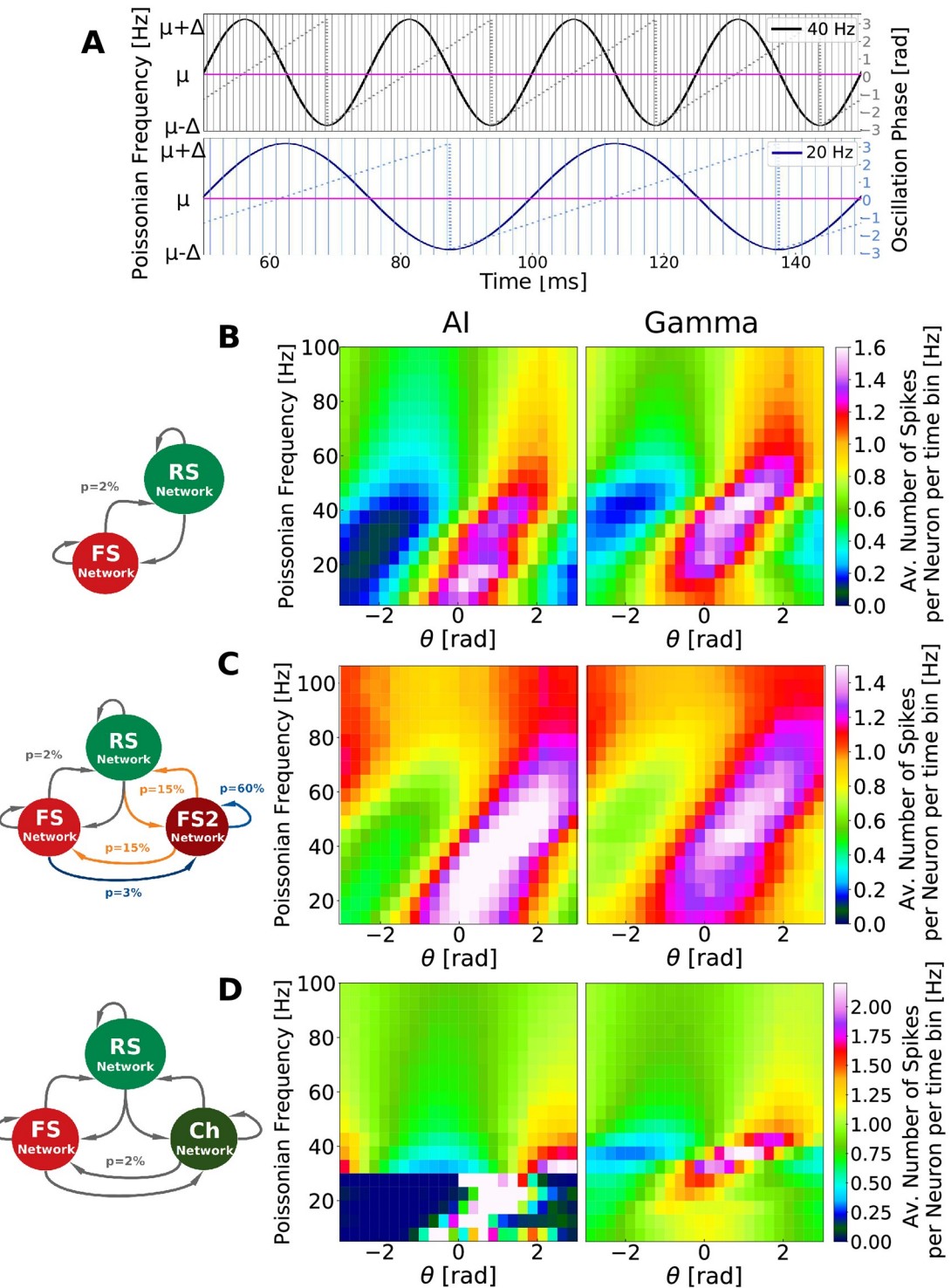

**Fig 9. Resonant properties of computational models.** A: Representation of external Poissonian noise varying in time in a sinusoidal manner around $\mu_{noise}$. In this protocol sinusoidal frequencies varied from 5 Hz to 100 Hz (step of 5 Hz). Two oscillatory frequencies are depicted: 20Hz (blue) and 40 Hz (black), together with their phases (second axis) and time bins (vertical line). For all frequencies the average Poissonian noise ($\mu_{noise}$) was kept the same, varying from $\mu_{noise} - \Delta_{noise}$ and $\mu_{noise} + \Delta_{noise}$. The bins were chosen in a way in which the oscillatory phases (from $-\pi$ to $\pi$) were divided into 25 intervals (for all frequencies), resulting in time bins of different duration for each oscillatory frequency. B: Resonant properties of *PING* Network. C: Resonant properties of *ING* Network. D: Resonant properties of *CHING* Network. The color maps displayed in B, C and D depict, for each oscillatory frequency and oscillation phase, the average number of spikes per RS neuron per time bin, during Gamma and AI-like states. All

values were normalized by the average firing inside of each state to exclude the state dependent firing rate level (which is higher on Gamma). $\Delta_{noise}$ = 0.5 Hz in all network models but $\mu_{noise}$ varied in each case. For AI, in PING and ING Networks $\mu_{noise}$ = 2 Hz and in CHING Network $\mu_{noise}$ = 1 Hz, while for Gamma, $\mu_{noise}$ = 3 Hz in in PING and ING Networks and $\mu_{noise}$ = 2 Hz in CHING Network.

time as well as their phase-preference. The analysis performed on this work is very qualitative, since it was based on a single patient. Nonetheless, this very sparse participation of RS and FS cells during Gamma was seen in different patients, and the same was observed in monkey for beta oscillations [21].

## Responsiveness

The occurrence of Gamma rhythms have been correlated with conscious perception [77–81] and several authors support these rhythms as being a suitable marker of consciousness. On the other hand, it has been proposed that the Asynchronous and Irregular activity, observed during awake and aroused states, due to its specific responsiveness properties, is an ideal setting for integrating multiple external inputs [12]. In support of this, it was concluded in a review that asynchronous states constitute the most reliable correlate of conscious states [82].

Previous work [12] has compared the responsiveness of a fully synchronized network (spike-to-spike regime) with a network in AI state, showing that the AI state is the best state to integrate multiple external inputs. It was also shown that, in rate-based networks, the most chaotic states could display the highest responsiveness, as measured using Shannon information [83]. In the present work, we compared the responsive properties of AI state with Gamma states generated by means of three different mechanism: PING, ING and CHING. Each of these networks were submitted to two types of inputs. First, a slowly-varying input integrated by the population of neurons over a substantial period of time (*integrative mode*). Second, we examined precisely-timed inputs, occurring in a time window smaller than the Gamma period (*coincidence detection mode*). For the *integrative mode*, we systematically found that the Gamma oscillations yielded less responsiveness than the AI-like states and even lesser responsiveness than real AI states (generated by the *AI Network*, used as a control). In the *coincidence detection mode*, we found that the response was only weakly modulated by the phase of the Gamma. This was assessed by comparing the Gamma oscillation to a sinusoidal control input, in which case the response was clearly phase-dependent. In agreement with the *integrative mode*, the responsiveness measured in the *coincidence detection mode* protocol was generally higher for the AI-like states. In addition, in the *coincidence detection mode*, among the three models, the *ING Network* is the only one that presents a similar responsiveness between Gamma and AI states, which stresses again the importance of network topology on networks behaviors.

A smaller responsiveness during Gamma states is somehow surprising since neurons are in general more depolarized in this state and additionally increase their firing, as we showed in our data analysis. On the other hand this observation is intuitively easy to understand, if we take into account the fact that Gamma oscillation are composed of successions of periods of high inhibition, which define time windows in which neurons are less likely to spike. While during Gamma states, these time windows of high inhibition constrain the times a certain neuron can spike, during AI states neurons can spike at all moments with the same probability. Indeed, we observed that the response during Gamma oscillations is phase-dependent, while there is no phase preference during AI states. However, although there was a phase dependence, Gamma oscillations did not provide a preferred phase where the network is more responsive than during AI states. The fact that higher levels of inhibition during Gamma could

explain their diminished responsiveness should be testable experimentally using intracellular recordings *in vivo*.

Given our model results, what this decrease of responsiveness could be useful for, and what are the advantages of a higher responsive state in AI? This questions can be approached in the light of the *Phase Coding Theory* (PC). This theory was initially formulated with respect to Theta rhythm [6], but lately extended to Gamma [7]. This theory states that, within the Gamma cycle, the excitatory input to pyramidal cells is converted into a temporal code whereby the amplitude of excitation is re-coded in the time of occurrence of output spikes relative to the cycle [7]. In this view, the cells that are most excited fire earlier in the cycle, while cells that are not excited enough are prohibited to spike due the new wave of inhibition composing the cycle. This process can be seen as a winner-take-all phenomena (or more precisely a few-winners-take-all phenomena, since it involves several neurons neurons) [7]. Such a coding strategy enables transmission and read out of amplitude information within a single Gamma cycle without requiring rate integration, proving a fast processing and readout by means of coincidence detection, rather than on rate integration [84], in agreement with more recent work [60, 76, 85]. Furthermore, this type of encoding strategy would, in principle, allow an improvement of signal-to-noise ratios, since neurons not conveying information would be hindered to spike. In this perspective, according to our models, Gamma oscillations would allow a network to respond quicker at the expense of decreasing the strength of its response. On the other hand, more responsive states such as AI, would be better suited to respond to low amplitude stimulus (due to their high sensitivity) at the cost of loosing temporal precision. Thus, AI states, because of their high responsiveness, seem well suited to detect inputs, while gamma oscillations, due to their tighter time precision, seem better suited to transmit timing information. Such possibilities constitute interesting directions to explore by future models.

## Resonance

In this work we reproduced previous results [75] showing that resonance is a fundamental property of spiking networks composed of excitatory and inhibitory neurons. We compared the resonant properties during AI and Gamma states generated by three different mechanism (ING, PING and CHING) and verified that, apart from a shift on the resonant frequency center, the resonant properties around the Gamma band in all networks did not change significantly from one state to the other. We call the reader attention to the particularities of each network model, especially the enlarged potential of resonance of ING network during AI.

Even though previous work proposed the importance of resonance in information transfer and processing in the brain [10], this aspect has been left aside until recently [86]. The most popular view, known as the Communication Through Coherence (CTC) Theory [8, 9], proposes a mechanistic explanation for how different neural regions could communicate by means of *coherence* [60]. This theory advocates that, since oscillations generate a rhythmic modulations in neuronal excitability (defining time windows in which neurons are capable to respond), only coherently oscillating groups can effectively communicate. In contrast, a recent work [86] present results indicating that, to the contrary, *coherence* is a consequence of communication, not a cause of it. This study shows that if an oscillating network is connected to another network that owns resonant properties around this same frequency, these two networks present *coherent* activity, and that the presence of these resonant interactions could explain more than 50% of the observed *coherence*. Furthermore, they show that the oscillating network sends information to the resonant one (the *Granger-causality* between field potentials is dominated by oscillatory synchronization in the sending area).

In this perspective, the enlarged potential of resonance of ING network in different bands during AI, indicates that this type of network structure (with heterogeneous connectivity patterns in between inhibitory neurons) could potentially convey information equally well in several bands. This stress the importance of network topology for neuronal information processing and also constitutes interesting directions to further explore.

## Supporting information

**S1 Fig. Synaptic time scale parameter search of a network composed of RS and FS neurons randomly connected.** The network used to produce this figure was composed of 20000 excitatory Regular Spiking and 5000 inhibitory Fast Spiking neurons connected randomly with a probability of connection of 2%. All synapses were delayed by a time delay of 1.5 ms, and had reference synaptic strengths of $Q_e^R = 1$ nS or $Q_i^R = 5$ nS and reference synaptic time scales of $\tau_e^R = \tau_i^R = 5$ ms. Synaptic strengths ($Q_{e,i}$) were normalized at each tested time scale ($\tau_{e,i}$) to keep the same synaptic gain, such that: $Q_{e,i} = (Q_{e,i}^R . \tau_{e,i}^R)/\tau_{e,i}$. A: Network oscillation frequency depicted in a color scheme as a function of excitatory and inhibitory synaptic time scales. White color corresponds to regions in which no oscillation was identified in RS population. B: Synchrony Index of RS population (top) and network balance (bottom) as a function of synaptic time scales. The Synchrony Index (SI) is based on the auto-correlation of the population frequency of RS cells. To be calculated, the autocorrelation of the population frequency was fitted by a damped cosine function and the value of this fitted function at zero time lag was defined as the SI. If the exponential decay rate was higher then 100, it was considered that there was no global oscillation at the population scale. The network balance was defined as the rate between the average excitatory and inhibitory synaptic currents, $\langle \frac{\langle I_{exc} \rangle_N}{\langle I_{inh} \rangle_N} \rangle_t$, in which $\langle \rangle_N$ stand for average among neurons and $\langle \rangle_t$ average on time. White squares indicate the two different parameter sets used in our simulations ($\tau_e = \tau_i = 5$ ms for *AI Network*, and $\tau_e = 1$ ms, $\tau_i = 7.5$ ms for *PING Network*). C: Same as B but calculated for the FS population. D: Population frequency autocorrelation of RS (green dots) and FS population (blue dots) neurons of the two used parameter sets. Solid lines indicate the damped cosine fitted function. (TIF)

**S2 Fig. *Gamma Network* parameter search.** The network connectivity ($p$) vs. inhibitory synaptic strengths ($Q_i$) parameter space of the Gamma Network are displayed as color-plots. A: Average spiking frequency. B: Network oscillation frequency. C: Network balance: rate between the average excitatory and inhibitory synaptic currents, $\langle \frac{\langle I_{exc} \rangle_N}{\langle I_{Inh} \rangle_N} \rangle_t$, in which $\langle \rangle_N$ stand for average among neurons and $\langle \rangle_t$ average on time. D: Membrane Potential Synchrony ($\chi$), calculated by means of the equation: $\chi^2 = \frac{\sigma_V^2}{\frac{1}{N}\Sigma_i^N \sigma_{V_i}^2}$, in which $V(t) = \frac{1}{N}\Sigma_i^N V_i(t)$, $\sigma_V^2 = \langle [V(t)]^2 \rangle_t - [\langle V(t) \rangle_t]^2$ and $\sigma_{V_i}^2 = \langle [V_i(t)]^2 \rangle_t - [\langle V_i(t) \rangle_t]^2$. The set of parameter which allowed Gamma Network to oscillate in the Gamma range are indicated by a star symbol. The white and yellow curves depict parameter choices in which the product between $p$ and $Q_i$ are the same. The yellow curve indicates all parameters equivalent to a choice of $p = 60\%$ and $Q_i = 5$ nS ($Q_i' = 3/p'$), while the white curve indicates all parameters equivalent to a choice of $p = 10\%$ and $Q_i = 5$ nS ($Q_i' = 0.5/p'$), like it is usually used in other works [30]. Every point in each graph is given by the average output of 10 simulations of 5 seconds each. In this simulations each neuron of the *Gamma Network* received 400 independent and identically distributed excitatory Poissonian spike trains with a spiking frequency $\mu_{Ext} = 5$ Hz and a synaptic strength of $Q_{Ext} = 1$ nS that decayed with synaptic time constant of $\tau_E = 5$ ms. E: Network activity for the parameters indicated with a start in A, B, C and ($p = 60\%$ and $Q_i = 5$ nS). The raster

plot of the whole network (e1), the population frequency (e2), the membrane potential of 3 randomly chosen neurons (e3) and the power spectrum of the population frequency (e4) are indicated. The population frequency is calculated as the total number of spikes (spikes of the whole network) in a time bin of 1 ms, divided by the duration of this time bin. Because of the exclusive presence of inhibitory neurons and its high level of recurrent inhibition, this network is capable of generating Gamma rhythms with frequencies around 70Hz by means of an ING mechanism.
(TIF)

**S3 Fig. Phase-locking statistical test.** A and B: Phase distribution of two randomly picked cells from the human recordings (Data segment 1): one excitatory (A, green) and one inhibitory (B, red). The phase distribution of each cell was fitted to a Von Mises curve, which allowed the estimation of its preferred phase $\overline{\theta_{VM}}$. The phase distribution of each neuron was tested for circular uniformity using a Bonferroni-corrected Rayleigh test [36, 37]. C and D: Rayleigh Z calculated for all recorded neurons: excitatory (C, green) and inhibitory (D, red). A neuron was considered phase-locked if the circular uniformity at P < 0.01, ($Z > Z_c$) could be rejected. In these plots, neurons were ordered according to their Z value and not according to their original indexes. E: Preferred phases, $\overline{\theta_{VM}}$, of each phase-locked cell, displayed in polar graph representation. Dark colored vectors indicate the average phase among each neuron type and $\Delta\theta$ the phase difference among RS and FS. Data segment 1 presented 22 minutes of recordings, containing 9 seconds of Gamma activity.
(TIF)

**S4 Fig. Firing rate change statistical test.** A: Activity of two randomly picked cells during several Gamma bursts: neuron 13 (inhibitory, left) and neuron 75 (excitatory, right). The graphs display the firing patter around Gamma bursts (indicated by the black doted lines). Each point corresponds to one spike in the correspondent tuple of time and burst ID (y-axis). B: Histogram computing the distributions of all spikes inside all Gamma bursts of neuron 13 (left) and neuron 75 (right). C: Exemplification of firing rate change statistical test. The Poissonian distribution of these two neurons is constructed based on their average firing rate calculated outside of Gamma bursts. The critical number of spikes $n_c$, indicated by the dotted lines, is calculated based on the *Percent Point Function* of the respective Poissonian Distribution for a period T, with an 95% Interval of Confidence. The observed number of spikes $n_{obsv}$ is depict as a dot over the curve. According to this procedure, only neuron 75 is considered to increase its firing, since $n_{obsv} > n_c$.
(TIF)

**S5 Fig. Phase preference of phase-locked cells per data segment in the human recordings.** A: Data segment 1—containing 22 minutes of recordings and 9 seconds of total Gamma activity. B: Data segment 2—containing 43 minutes of recordings and 14 seconds of total Gamma activity. C: Data segment 3—containing 28 minutes of recordings and 16 seconds of total Gamma activity. D: Data segment 4—containing 26 minutes of recordings and 13 seconds of total Gamma activity. E: Data segment 5—containing 16 minutes of recordings and 11 seconds of total Gamma activity. The preferred phases of each phase-locked cell are displayed in polar graph representation. Phases were calculated from $-\pi$ to $\pi$. The vector size gives a measure of the phase distribution of each cell. Big amplitude vectors indicate very concentrated distributions while small amplitude vectors indicate less concentrated ones. The color of each vector encodes the type of the cell of whom it represents the phase: red (FS), and green (RS). Cell number IDs are indicated. Dark colored vectors indicate the

average phase among each neuron type and $\Delta\theta$ the phase difference among them.
(TIF)

**S6 Fig. Change of *Gamma participating cells* with time in experimental data.** The middle panel represents each cell by a circle in each of the 5 data segments. FS and RS phase-locked cells are depicted respectively as red and green circles, while not phase-locked or inconclusive (with respect to phase locked) cells of both types are depicted as blue and gray circles respectively. Superposed to each cell circle, pointing up and down triangles were added to indicate if the cell increased ($\triangle$) or decreased ($\triangledown$) its firing. If the cell didn't change its firing significantly a minus sign (-) was added. Side box plots indicate, on the left, the percentage of phase-locked FS (red) and RS (green) cells in each of the 5 data segments, and, on the right, the percentage of firing rate increase. Dotted lines indicate the average value (phase-locking level: left and firing rate increase: right) between the 5 data segments. The bottom box plot depicts the superposed counts of phase-locking or firing rate increase behavior of each individual cell, computed in the 5 data segments.
(TIF)

**S7 Fig. Behavior consistency of RS and FS cells in human recordings.** Distributions of consistency indexes among the recorded neurons with respect to to firing rate increase are displayed respectively in A and B for RS cells and FS cells, while C and D display the consistency indexes distribution of phase-locking for RS and FS.
(TIF)

**S8 Fig. Neural behavior time distribution in the human data.** The activity of each neuron inside and outside Gamma bursts in all 5 data segments were quantified. Taking into account that each data segment had a different duration, containing a different total Gamma duration, and that some neurons were silent in some data segments, each neuron was analyzed individually, taking into account the percentage of the total amount of time in which the neuron was active. A: Phase-locking time distribution. The grid plot in the middle displays the amount of time (with respect to the total recording time) in which each neuron was considered phase-locked (A, y axis), and the the amount of time in which each neuron was considered not phase-locked (A, x axis). RS neurons are depicted in green and FS neurons in red, together with their ID number. Neurons lying outside of the diagonal are neurons of whom statistical analysis was inconclusive at some data segments, due to the reduced number of spikes. At the top left corner, lie neurons that were always considered phase-locked, while neurons that were never considered phase-locked are placed at the bottom right corner. Pie plots indicate the percentage of neurons that passed at least 50% of the total time being either phase-locked or not phase-locked (neurons that fall inside of the colored quadrants) and the neurons lying on the left white quadrant. B: Same analysis as A but displaying the firing rate change time distribution. This analysis indicates that only a small percentage of neurons passed at least 50% of the total time being either phase-locked (RS: 4.4%, FS: 13%) or increasing its firing (RS: 20.6%, FS: 52.2%). Moreover, even though no cell was 100% of the time phase-locked to Gamma, some cells were 100% of the time not phase-locked to Gamma (RS: 22.1%, FS: 13%) and others never increased their firing (RS: 41.2%, FS: 17.4%).
(TIF)

**S9 Fig. Firing rate distribution of individual neurons inside Gamma bursts and their phase-locking classification in human data recordings.** The average firing rate of each neuron in each of the 5 data segments (inside Gamma bursts) is depicted as a point in this graph (each neuron presents 5 points). The color of each point corresponds to the neuron classification with respect to phase-locking in the correspondent data segment (purple: phase-locked,

red: not phase-locked and gray: inconclusive). The average firing rate inside Gamma bursts was calculated based on the total Gamma duration (recorded by the electrode, that also recorded the particular neuron, in the respective data segment) and the total number of spikes emitted by this particular neuron exclusively inside the Gamma bursts of the respective data segment. Cells classified as inconclusive are cells that spiked less then 5 times inside Gamma bursts, or cells whose electrode measured less then 1 second of Gamma bursts in the respective data segment. FS neurons are depicted on the left and RS neurons on the right. Box plots referent to each neuron distribution are added to help in the visualization (regardless of the reduced number of points). The box extends from the lower to upper quartile values of the data, with a line at the median. The whiskers extend from the box to show the range of the data. Flier points are those past the end of the whiskers and are depicted with black circle together with the color point. This graph illustrates the fact that phase-locking and not phase-locking behaviors are observed both in cells with high and low firing rates.
(TIF)

**S10 Fig. Firing rate distributions.** Firing rate distributions of different neuron types (inside and outside Gamma bursts) are depicted in A, B, C and D for each studied system. A: Human recordings. B: *PING Network*. C: *ING Network* and D: *CHING Network*. Average firing rates of each cell type is indicated by the dotted line.
(TIF)

**S11 Fig. Average excitatory and inhibitory synaptic conductances.** A: Illustration of the analyzed system: *PING Network*, *ING Network* and *CHING Network*. B: Ratio between excitatory conductance ($G_e$) and leakage conductance ($G_L$). C: Ratio between inhibitory conductance ($G_i$) and leakage conductance ($G_L$). Averages are indicated by the dotted line. The distributions fall inside of the physiological range observed experimentally [87].
(TIF)

**S12 Fig. Average level of phase-locking.** The average level of phase-locking is defined as the averaged percentage of cells in the network considered to be phase-locked, across the 5 segments of data recorded. The analysis was done separately for excitation and inhibition. A: Human Data recordings, B:*PING Network*, C: *ING Network* and D: *CHING Network*. The percentage of cells signaled as *inconclusive* relates to cells in which the number of spikes inside Gamma burst were too small to allow statistical significant phase-locking.
(TIF)

**S13 Fig. Network responsiveness of a network composed of RS and FS neurons randomly connected with different synaptic time scales.** A: *AI network* receiving a Poissonian drive of 3Hz. B: *PING network* receiving a Poissonian drive of 3Hz (inducing Gamma). C: *PING network* receiving a Poissonian drive of 2Hz (not inducing Gamma). In addition to the drive each network received a Gaussian stimulus of 2Hz pick and a standard deviation of 50 ms. The drive and stimulus are depicted in each case in a1, b1 and c1. The raster plot of each network during the stimulation is depicted in each case in a2, b2 and c2. The membrane potential of 3 randomly picked neurons are depicted in each case in a3, b3 and c3. The raw and the filtered simulated LFP are depicted in each case in a4, b4 and c4.
(TIF)

**S14 Fig. Responsiveness of individual cells in computational models.** A: *PING Network*. B: *ING Network*. C: *CHING Network*. To estimate the individual cell responsiveness, we calculated the average spiking frequency of each cell inside (y-axis) and outside stimulus (x-axis) during AI-like states (left) and Gamma states (right). RS cells are displayed in green and FS

cells in red. In each plot the linear regression from the points is depicted with the identity. We observe that all cells follow the same rule of responsiveness (proportional to their firing outside the stimulus). No difference can be seen between the responsiveness of neurons classified as *Gamma participating* and the *Gamma non-participating* cells.
(TIF)

**S15 Fig. Phase-dependent network response protocol.** A: Protocol scheme in ING Network when it displays Gamma oscillations (45-65 Hz). Top: stimulus used to measure network phase-dependent response. The stimulus consisted of fast Gaussian fluctuation (standard deviation of 1 ms) which modulated the firing rate of the external Poissonian spike trains injected into network from 0 to 50 Hz. Middle: Raster plot indicating the network response to the Gaussian stimulus. The network responsiveness was calculated according to Eq 6, in a time window $T$ = 18ms (shaded gray area). Bottom: Gamma oscillation phase around the the stimulus pick. The phase at the time the stimulus was applied is indicated. The Phase-dependent network responsiveness was measured in three different network states: B: AI state (Poissonian noise = 2Hz, no external current). C: AI-modulated states (Poissonian noise = 1Hz, with sinusoidal external current). D: Gamma state (Poissonian noise = 3Hz, no external current). Items A, B and C display the Raster activity of ING Network without the Gaussian stimulation. Only 20% of network is shown.
(TIF)

**S16 Fig. Linear representation of color maps depicted in Fig 9.** A: Resonant properties of *PING* Network. B: Resonant properties of *ING* Network. C: Resonant properties of *CHING* Network. The curves displayed in B, C and D depict, for each oscillatory frequency (color scheme) the amplitude (average number of spikes per neuron per time bin) as a function of the oscillation phase, during Gamma and AI-like states. All values were normalized by the average firing inside of each state to exclude the state dependent firing rate level (which is higher on Gamma). $\Delta_{noise}$ = 0.5 Hz in all network models but $\mu_{noise}$ varied in each case. For AI, in PING and ING Networks $\mu_{noise}$ = 2 Hz and in CHING Network $\mu_{noise}$ = 1 Hz, while for Gamma, $\mu_{noise}$ = 3 Hz in in PING and ING Networks and $\mu_{noise}$ = 2 Hz in CHING Network.
(TIF)

**S17 Fig. Resonant properties of computational models during Gamma in each cell type.** A: Resonant properties of *PING* Network. B: Resonant properties of *ING* Network. C: Resonant properties of *CHING* Network. The color maps displayed in A, B and C depict, for each oscillatory frequency and oscillation phase, the average number of spikes per cell type (RS, FS, FS2 or Ch) and time bin, during Gamma state. Differently than Fig 9 no normalization was applied. $\Delta_{noise}$ = 0.5 Hz in all network models but $\mu_{noise}$ varied in each case. In PING and ING Networks $\mu_{noise}$ = 3 Hz and in CHING Network $\mu_{noise}$ = 2 Hz.
(TIF)

# Acknowledgments

We thank Damien Depannemaecker and Mallory Carlu for enlightening discussions and insights.

# Author Contributions

**Conceptualization:** Eduarda Susin, Alain Destexhe.

**Data curation:** Eduarda Susin.

**Formal analysis:** Eduarda Susin.

**Funding acquisition:** Eduarda Susin, Alain Destexhe.

**Investigation:** Eduarda Susin.

**Methodology:** Eduarda Susin, Alain Destexhe.

**Project administration:** Eduarda Susin, Alain Destexhe.

**Resources:** Alain Destexhe.

**Software:** Eduarda Susin.

**Supervision:** Alain Destexhe.

**Visualization:** Eduarda Susin.

**Writing – original draft:** Eduarda Susin.

**Writing – review & editing:** Alain Destexhe.

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
