## [Decision Letter · Decision Letter 0]

23 Jun 2021

Dear Mrs Susin,

Thank you very much for submitting your manuscript "Integration, coincidence detection and resonance in networks of spiking neurons expressing gamma oscillations and asynchronous states" for consideration at PLOS Computational Biology.

As with all papers reviewed by the journal, your manuscript was reviewed by members of the editorial board and by several independent reviewers. In light of the reviews (below this email), we would like to invite the resubmission of a significantly-revised version that takes into account the reviewers' comments.

We cannot make any decision about publication until we have seen the revised manuscript and your response to the reviewers' comments. Your revised manuscript is also likely to be sent to reviewers for further evaluation.

Sincerely,

Arvind Kumar, Ph.D.

Guest Editor

PLOS Computational Biology

Samuel Gershman

Deputy Editor

PLOS Computational Biology

Reviewer's Responses to Questions

**Comments to the Authors:**

Reviewer #1: The manuscript describes computer simulations of 3 neural network models. The authors compare the dynamics of their models during gamma oscillation bursts to some previously published human recordings data and then test the model activity responses under different external stimuli. The simulations appear to have been competently performed. However, I do not find particularly striking the main conclusion that the networks are less responsive during gamma oscillation bursts. It should in any case be made more convincing (point 4 below).

At the technical level, several points should be addressed :

-1) In the comparison between the different networks and the data, I did not find the value of spiking rates of the FS2 neurons (the only information that I found is Fig S1A where it appears that FS2 cells fire at a very low frequency -in this figure Panel A and B seems to be interverted as compared to the caption). I would be helpful to give it, for instance to give separately FS and FS2 histograms in Fig S7).

-2) A related point is that an oscillation of 50Hz appears quite low for an ING network. For instance, given the results of ref.36, one would expect a pure ING to be more suitable for the description of 100-200Hz oscillation. Is this related to the low firing rate of the FS2 neurons?

How sensitive is this oscillation frequency? Comments from the authors on this point would be appreciated.

-3). I am somewhat confused by the authors’ convention and discussion of the relative firing phases and order of the different populations I the oscillations. I would expect the phase to be proportional to time, so that a population with the larger phase fires after one with a smaller phase i.e. that time would go in counterclockwise direction in the circular representation of Fig 6. For instanc, in Fig.6 from the diagram I would expect the FS population (red) to fire before the RS one (green). However, reading from the text, the converse is described by the authors. Please clarify.

-4) That the network are less sensitive to external inputs during gamma bursts appears to be the central result of the paper. However, the external input is provided on top of the input creating the gamma burst. Thus, one wonders whether this diminishes sensitivity is due to the gamma oscillation or simply to the nonlinear addition of two inputs. It would be good to compare with a parameter case in the AI regime where the first input does not create oscillation to trry and discriminate the two effects.

-5) I do not find particularly clear or precise (e.g. it is difficult to distinguish the different shades of green in C right) the depiction of resonance in Fig.9. Plotting the resonance curve amplitude and phase as a function of frequency, as usually done, would seem less colorful but less demanding on the reader’s part.

-6) There are several minor english mistakes that should be corrected (“an stimulus” multiple times, “cells…had its” line 243, “participates of” line 268, odd sentence line 268-269, “an structure” line 299,…)

Reviewer #2: The authors have analyzed the human electrophysiology data and studied the properties of LFP gamma oscillations and the spiking activity of the single cells. They have shown that only a small percentage of the recorded neurons are phase-locked to the gamma oscillations and show elevated activity during the gamma bursts. They have then developed three different spiking neuronal models that could produce asynchronous activity as well as gamma oscillations through different mechanisms and have explored in what extent they can produce the same properties observed in experimental data. They have also studied the response of the three networks to different types of external stimuli and have concluded that the in all the three networks, larger response is observed when they are in irregular-asynchronous state.

I think the manuscript provides interesting results and can be considered as an important contribution in the context of the dynamics of cortical networks. I just have several comments that might be useful to increase the readability and coherency of the manuscript.

General: Do the authors implicitly assumed that the gamma bursts emerge through mean input augmentation in Human subject, like the model? This could not always be the case since even with homogeneous input rate in a certain range, the gamma oscillations could spontaneously wax and wane and show quite variable amplitude like what is seen in Ref [82] of the manuscript. I think the authors need to elaborate this point.

General: the results show that responsiveness of the simulated networks to external inputs, either pulse packets or rate modulation, is higher at AI state compared with the oscillatory state.

However these results could not be readily used to argue in favor of the superiority of the AI state. That is because although the irregular dynamics is advantageous in responsiveness, the signals and information are better transmitted in oscillatory networks (Akam and Kullmann, Neuron 2010; Sherfey et al., Neurobiology of Learning and Memory 2020). I think presence of such a compromise between responsiveness and transmission should be discussed somewhere in the manuscript.

Line 14: “biding”, correct

Line 21: weak pairwise correlations can also be case for the synchronous state when the neurons sparsely fire. Please restate the AI state more precisely.

Lines 30 and 31: I think it is adequate to bring references here; although later in the results the references are introduced.

Lines 70 to 76: with this arrangement it seems that the inputs to the neurons have a low degree of correlation that certainly affects the dynamics and make it different with the case where the inputs are completely uncorrelated. Can the authors quantify this correlation or at least indicate the presence of such correlation?

Line 81: for me the decay time of the excitatory connections are smaller than those usually used in other references. Can the authors justify their choice?

Line 81: orand ?

Line 83: “were computed inside of the synaptic current term”. I think this can be reworded.

Lines 94 to 96: again the choice of the decay times is not justified. Do the authors implicitly mean that for an AI state the synaptic decay times should be equal? A reference or more clarification might help here.

General in methods: It seems that subscript for “external” is used with capital initials in some cases and lowercase in others. Please make it coherent.

Lines 144 to 147: I think more details should be brought for how LFP is calculated.

Lines 148 to 159: definition of the gamma bursts in the experiments is ambiguous to me. First, I see that in many cases gamma oscillations are present outside the gamma burst and since the comparison is made with the baseline average, the outcome depends on if the baseline is itself an oscillatory state or is an asynchronous one. Moreover, in experimental study with multiple electrodes (referring to Fig 1A), it seems that the gamma burst can be observed in different times for different electrodes, or simultaneously for all the electrodes. This latter leads to the question that which is considered in the calculation of ~13 ms of the total time of gamma bursts in each recording session. I hope that the authors can make this point more clear.

For example, one can think that if the criterion for the gamma bursts is moderated, how the results change? Or if the criterion is chosen an absolute one instead of a differential one?

Line 154: could a reference be added for Keiser filter?

Lines 169 and 170: check if the capital are needed.

Lines 239 to 241: how this result can be inferred from the figures?

Line 248: I think that a major un-addressed point is that if the neurons that show higher rate in the gamma bursts are those that show higher phase locking? Could this analysis be made or could it be discussed?

Fig 2: please indicate the shaded region in panel B and C are, presumably, the burst period. Moreover, I suggest to increase the size of panel A or only show number of sample electrodes for more clarity.

Fig 4 and general: some legends of the figures are too small and cannot be read when the size of paper is standard. Please recheck all the figures.

Line 345: “Fig 5 illustrates how network heterogeneities in network connections (ING Network) or in neuron types (CHING Network) influence network activity.” I understand what the authors meant but the sentence gives a sense that the effect of heterogeneity is systematically studied in the figure. Please moderate the expression.

Line 370: weaker than?

Line 418: “a restrict group” please correct.

Line 723: “It had an amplitude of 50Hz” could be reworded, and meddle->middle?

**Have the authors made all data and (if applicable) computational code underlying the findings in their manuscript fully available?**

Reviewer #1: Yes

Reviewer #2: Yes

PLOS authors have the option to publish the peer review history of their article (what does this mean?). If published, this will include your full peer review and any attached files.

Reviewer #1: No

Reviewer #2: **Yes: **Alireza Valizadeh
---

## [Decision Letter · Decision Letter 1]

2 Sep 2021

Dear Mrs. Susin,

We are pleased to inform you that your manuscript 'Integration, coincidence detection and resonance in networks of spiking neurons expressing gamma oscillations and asynchronous states' has been provisionally accepted for publication in PLOS Computational Biology.

Best regards,

Arvind Kumar, Ph.D.

Guest Editor

PLOS Computational Biology

Samuel Gershman

Deputy Editor

PLOS Computational Biology

I am pleased to inform you that now the two reviewers are happy with the current version of the manuscript. And now the manuscript can proceed to production cycle. Congratulations!!

Reviewer's Responses to Questions

**Comments to the Authors:**

Reviewer #1: The authors have considered my comments and taken them into account.

Minor point: I do not understand the statement of the authors saying that when phases are measured from 0 to 2 pi, time and phase are proportional but because they measure angles from -pi to pi, time and phase run in opposite ways (I understand that since phases are periodic it is difficult to say that one is before the other (usually one considers the smallest absolute difference between the phases of the two processes with suitable addition of mutiple of 2pi which is I believe what the authors do). With the usual convention, phases grow with time -ie rotate counterclockwise in the polar plot- so the largest one hit a given value first and is in advance of the smaller one (my apologies since I may have created the confusion with the question in my previous report).

Reviewer #2: The Authors have adequately addressed the comments and I think the manuscript can be accepted for publication.

**Have the authors made all data and (if applicable) computational code underlying the findings in their manuscript fully available?**

Reviewer #1: Yes

Reviewer #2: None

PLOS authors have the option to publish the peer review history of their article (what does this mean?). If published, this will include your full peer review and any attached files.

Reviewer #1: No

Reviewer #2: **Yes: **Alireza Valizadeh

---

## [Editor Report · Acceptance letter]

13 Sep 2021

PCOMPBIOL-D-21-00829R1 

Integration, coincidence detection and resonance in networks of spiking neurons expressing gamma oscillations and asynchronous states

Dear Dr Susin,

I am pleased to inform you that your manuscript has been formally accepted for publication in PLOS Computational Biology. Your manuscript is now with our production department and you will be notified of the publication date in due course.

With kind regards,

Andrea Szabo
